# Computing Optimal Regularizers for Online Linear Optimization

## Abstract

Follow-the-Regularized-Leader (FTRL) algorithms are a popular class of learning algorithms for online linear optimization (OLO) that guarantee sub-linear regret, but the choice of regularizer can significantly impact dimension-dependent factors in the regret bound. We present an algorithm that takes as input convex and symmetric action sets and loss sets for a specific OLO instance, and outputs a regularizer such that running FTRL with this regularizer guarantees regret within a universal constant factor of the best possible regret bound. In particular, for any choice of (convex, symmetric) action set and loss set we prove that there exists an instantiation of FTRL which achieves regret within a constant factor of the best possible learning algorithm, strengthening the universality result of Srebro et al., 2011.

Our algorithm requires preprocessing time and space exponential in the dimension $d$ of the OLO instance, but can be run efficiently online assuming a membership and linear optimization oracle for the action and loss sets, respectively (and is fully polynomial time for the case of constant dimension $d$). We complement this with a lower bound showing that even deciding whether a given regularizer is $\alpha$-strongly-convex with respect to a given norm is NP-hard.

## 1 Introduction

Online Linear Optimization (OLO) is one of the most fundamental problems in the theory of online learning. Here, a learner must repeatedly (for $T$ rounds) select an action $x_t$ from some bounded convex action set $\mathcal{X}$. Simultaneously, an adversary selects a linear loss function $\ell_t$ from a bounded convex loss set $\mathcal{L}$, and the learner receives loss $\langle x_t, \ell_t \rangle$. The learner would like to minimize their total loss, and more specifically minimize their *regret*: the gap between their total loss and the loss of the best fixed action $x^* \in \mathcal{X}$ in hindsight.

By choosing the action set $\mathcal{X}$ and loss set $\mathcal{L}$ appropriately, online linear optimization captures many other learning-theoretic problems of interest. For example, when $\mathcal{X} = \Delta_d$ (distributions over $\{1, 2, \ldots, d\}$) and $\mathcal{L} = [0, 1]^d$, this captures the classical problem of *learning with experts*. Similarly, when the loss set $\mathcal{L}$ is the $\ell_2$ unit ball, this variant of OLO is the core subproblem involved in *online convex optimization* (specifically, of a Lipschitz function with domain $\mathcal{X}$). Even more generally, the works of Gordon et al. (2008) and Abernethy et al. (2011) demonstrate how to reduce the problems of linear $\phi$-regret minimization (including swap regret minimization) and Blackwell approachability to different instances of OLO. These problems in turn have many applications extending past learning theory, from designing algorithms for computing correlated equilibria in repeated games, to producing calibrated forecasts, to constructing classifiers satisfying a variety of fairness criteria (Farina et al., 2021; Okoroafor et al., 2024; Chzhen et al., 2021).

For this reason, it is an extremely relevant problem to understand the best possible regret bounds achievable for different instances of OLO. Here, the state-of-the-art leaves something to be desired. It is well-known that learning algorithms such as Follow-The-Regularized-Leader (FTRL) achieve regret that scales with $O(\sqrt{T})$, and that this dependence on $T$ is tight. However, the dependence of the optimal regret on the sets $\mathcal{X}$ and $\mathcal{L}$ (e.g., how the constant factor in the above regret bound depends on the dimension $d$ of these sets) is in general poorly understood.

Moreover, FTRL is not a single algorithm, but a family of algorithms parametrized by a convex function $f : \mathcal{X} \to \mathbb{R}$ called the *regularizer*. The actual regret bounds achieved by FTRL can vary greatly depending how the choice of regularizer interacts with the geometry of $\mathcal{X}$ and $\mathcal{L}$. For example, running FTRL with the quadratic regularizer results in an $O(\sqrt{dT})$ regret algorithm for the learning with experts problem; however, running FTRL with the negative entropy regularizer results in an algorithm with a tight $O(\sqrt{T \log d})$ regret bound, with an exponential improvement in dimension over the quadratic choice of regularizer. On the other hand, there exist other instances (choices of $\mathcal{X}$ and $\mathcal{L}$) where the quadratic regularizer is optimal. Understanding what the optimal choice of regularizer is for a given instance of OLO is a major open problem.

## 1.1 OUR CONTRIBUTIONS

For any action set $\mathcal{X}$ and loss set $\mathcal{L}$, the optimal possible regret bound (as $T$ goes to infinity) scales as $\mathrm{Rate}(\mathcal{X}, \mathcal{L})\sqrt{T} + o(\sqrt{T})$, for some constant $\mathrm{Rate}(\mathcal{X}, \mathcal{L})$. Our goal in this paper is to design learning algorithms which approximately achieve this optimal regret bound. Specifically, we want to algorithmically construct learning algorithms with worst-case regret at most $C \cdot \mathrm{Rate}(\mathcal{X}, \mathcal{L})\sqrt{T}$ for some universal constant $C$ that holds for any choice of action set and loss set in any dimension. For technical reasons, we restrict our attention in the following results to action sets $\mathcal{X}$ and loss sets $\mathcal{L}$ that are *centrally symmetric* – it is an interesting open direction to extend these results to fully general choices of $\mathcal{X}$ and $\mathcal{L}$.

We begin by showing that the optimal regret bound is achieved by some instantiation of Follow-The-Regularized-Leader. We do so by extending earlier work of Srebro et al. (2011) who, by analyzing the martingale types of Banach spaces, demonstrated that there is always an instance of FTRL which achieves regret $O(\mathrm{Rate}(\mathcal{X}, \mathcal{L})(\log T)\sqrt{T})$. In Theorem 7, we show that a more careful analysis of these martingale types allows us to remove this $\log T$ factor and prove that some variant of FTRL is within a universal constant of optimal.

Although the above argument proves the existence of a near-optimal instance of FTRL, it is highly non-constructive. In the remainder of the paper we study the following algorithmic question: given sets $\mathcal{X}$ and $\mathcal{L}$ (e.g., via oracle access), how can we compute the optimal regularizer for these sets? Ultimately, we provide an algorithm that takes as input $\mathcal{X}$ and $\mathcal{L}$ (via standard oracle access to both sets), runs in time $\exp(O(d^2 \log d))$, and outputs a regularizer $f$ with the property that the worst-case regret of FTRL with $f$ is at most a universal constant times $\mathrm{Rate}(\mathcal{X}, \mathcal{L})\sqrt{T}$ (Theorem 1).

The main technical ingredient in this algorithm is a new method for optimizing over the set of convex functions that are $\alpha$-strongly convex with respect to a given norm. This is important for the above problem because one can show that for any regularizer $f$, the regret of running FTRL with that regularizer is bounded by $O(\sqrt{D\alpha T})$ if the range of $f$ over $\mathcal{X}$ (the maximum value of $f$ minus the minimum value of $f$) is at most $D$ and if $f$ is $\alpha$-strongly-convex with respect to the norm induced by the dual set of the loss set $\mathcal{L}$. We can show that this regret-bound is constant-factor-optimal for the near-optimal variant of FTRL in Theorem 7, and hence it suffices to try to minimize $D\alpha$ over all convex functions $f$.

To do this, we first show that we can approximate any smooth convex function $f$ as a maximum of several "quasi-quadratic" functions: quadratic functions $g_{x_0}$ centered at some point $x_0$ with a small cubic term which guarantee that that the contribution of $g_{x_0}$ to the Hessian of $f$ decays far from $x_0$. Note that these are not just approximations of the values of $f$, but also also the gradients and Hessians of $f$; in particular, if the original function was $\alpha$-strongly-convex with respect to some norm, our approximation will be similarly strongly-convex.

By restricting our quasi-quadratic functions to be centered at points belonging to a (large but) finite discretization of $\mathcal{X}$, we demonstrate how to optimize over this set of approximations by solving a large convex program with variables for the values, gradients, and Hessians of the quasi-quadratic functions at each point in the discretization. Solving this convex program involves implementing a separation oracle to verify whether a specific approximation is $\alpha$-strongly-convex with respect to an arbitrary norm.

As stated earlier, this approach takes time exponential in the dimension of the action and loss sets (although is completely independent of the time horizon $T$, and thus efficient for constant dimension

$d$). We complement this with a lower bound showing that even verifying whether a regularizer $f$ is $\alpha$-strongly-convex at a specific point $x \in \mathcal{X}$ requires exponentially many oracle queries to $\mathcal{L}$.

## 2 RELATED WORK

**Applications of Online Linear Optimization.** The problem of Online Linear Optimization (and its generalization, Online Convex Optimization) are central problems in the field of online learning – we refer the reader to Hazan et al. (2016) for a general-purpose introduction. Traditionally OLO is studied in the case where the action sets and loss sets are unit balls in a standard norm (e.g. the $\ell_1$, $\ell_2$, or $\ell_\infty$ norms). However, there are many motivating settings where we wish to minimize regret with less standard sets. Several authors (Takimoto & Warmuth, 2003; Kalai & Vempala, 2005; Koolen et al., 2010; Audibert et al., 2014) study variants of OLO where the action space has some combinatorial structure – for example, $\mathcal{X}$ could be the spanning tree polytope, or the polytope formed by all $s$-$t$ paths in a graph. Minimizing external regret in extensive form games – one standard method for computing coarse correlated equilibria Farina et al. (2020) – involves solving an instance of OLO where $\mathcal{X}$ is the sequence form polytope. Finally, as mentioned earlier, the work of Abernethy et al. (2011) and Gordon et al. (2008) allows us to translate any instance of Blackwell approachability or $\phi$-regret minimziation to a (usually non-standard) instance of OLO.

**Follow-The-Regularized-Leader and Mirror Descent.** The Follow-The-Regularized-Leader algorithm can be thought of as a form of *mirror descent*, a family of first-order optimization algorithms that generalize gradient descent by using arbitrary distance-generating functions. Originally, mirror descent was proposed by Nemirovski & Yudin (1978) as an offline optimization algorithm with $\ell_p$ norm constraints and $\ell_q$ Lipschitz assumptions, and was shown to have minimax optimal query complexity. Sridharan & Tewari (2010) studied the optimality of mirror descent for online linear optimization when the action and loss vectors are in the unit ball of two Banach spaces dual to each other, proving the existence of a regularizer for mirror descent that almost achieves the minimax rate under an adaptive adversary. Later, Srebro et al. (2011) extended this approach to cases where the action and loss vectors come from independent convex balls in primal and dual Banach spaces. The existence of such strongly convex regularizers is also linked to the Burkholder method introduced by Foster et al. (2018) for more general online learning problems. In particular, the authors propose that given an online learning instance and a target regret bound, the existence of a Burkholder function for that instance guarantees the existence of a prediction strategy that achieves the desired regret. Notably, taking the dual of this Burkholder function for the online linear optimization (OLO) problem results in a strongly convex regularizer that can be used effectively with FTRL Foster et al. (2018).

Many modern learning algorithms are actually variants of mirror descent / FTRL (Block, 1962; Zinkevich, 2003; Kivinen & Warmuth, 1997; Littlestone, 1988; Kakade et al., 2010; Warmuth & Kuzmin, 2007). Recently, Jin & Sidford (2020) used a variant of mirror descent to solve infinite-horizon MDPs, achieving linear runtime in the number of samples. Aubin-Frankowski et al. (2022) extended mirror descent to optimize convex functionals on an infinitesimal space, demonstrating that the primal iterations of Sinkhorn's algorithm for entropic optimal transport in a continuous domain are an instance of mirror descent. Wibisono et al. (2022) studied alternating mirror descent for two-player bilinear zero-sum games, proving a regret bound of $O\left(T^{1/3}\right)$. Mirror descent has also been used in the context of stochastic optimization Nemirovski et al. (2009). Authors in Duchi et al. (2010) study mirror descent for composite loss functions under both stochastic and online settings. Lei & Tang (2018) relaxed the subgradient boundedness condition from Duchi et al. (2010) and extended their analysis to examine the generalization performance of multi-pass SGD in non-parametric settings. Dani et al. (2008); Cesa-Bianchi & Lugosi (2011); Bubeck et al. (2012) applied mirror descent to address the problem of online linear optimization with bandit feedback. Allen-Zhu & Orecchia (2014) introduced a novel interpretation of mirror descent as optimizing a dual-based lower bound for the objective. Building on this perspective, they proposed a coupling between mirror descent and gradient descent that achieves an accelerated convergence rate. (Yuan et al., 2020; Shahrampour & Jadbabaie, 2017) applied mirror descent in distributed settings. Lobos et al. (2021) utilized mirror descent for a constrained online revenue maximization problem with unknown parameters. Authors in (Bansal & Coester, 2021; Lu et al., 2020; Balseiro et al., 2023) employ mirror descent for online resource allocation problems. Mirror descent has also been instrumental

in primal-dual methods for solving structured saddle-point problems (Nesterov, 2009; Tiapkin & Gasnikov, 2022; Bayandina et al., 2018; Sherman, 2017; Jambulapati & Tian, 2024; Jambulapati et al., 2020).

# 3 PRELIMINARIES

## 3.1 ONLINE LINEAR OPTIMIZATION

We begin by defining the problem of *online linear optimization* (OLO). In this problem, every round $t$ (for a total of $T$ rounds) the learner must pick an action $x_t$ from a convex action set $\mathcal{X} \subset \mathbb{R}^d$. The adversary then picks a loss vector $\ell_t$ from a convex loss set $\mathcal{L}$, after which the learner suffers loss $\langle x_t, \ell_t \rangle$ and observes the loss vector $\ell_t$. The learner would like to minimize their total loss, and more specifically minimize their total *regret*: the gap between their loss and the loss of the best action in hindsight. Formally, given a sequence of learner actions $\mathbf{x} = (x_1, x_2, \ldots, x_T)$ and losses $\boldsymbol{\ell} = (\ell_1, \ell_2, \ldots, \ell_T)$, the regret of the learner is given by

$$\text{Reg}(\mathbf{x}, \boldsymbol{\ell}) = \sum_{t=1}^{T} \langle x_t, \ell_t \rangle - \sum_{t=1}^{T} \min_{x^* \in \mathcal{X}} \langle x^*, \ell_t \rangle.$$

The learner chooses their actions according to some learning algorithm $\mathcal{A}$, which can be thought of as a function $\mathcal{A}$ mapping a sequence of losses $\boldsymbol{\ell} = (\ell_1, \ell_2, \ldots, \ell_T)$ to a sequence of actions $\mathbf{x} = (x_1, x_2, \ldots, x_T)$ in such a way that $x_t$ depends only on the history of losses $\ell_1, \ell_2, \ldots, \ell_{t-1}$ until round $t-1$. We define the $T$-round regret $\text{Reg}_T(\mathcal{A})$ to be the worst-case regret suffered by algorithm $\mathcal{A}$ against an adversarially chosen sequence of losses, i.e., $\text{Reg}_T(\mathcal{A}) = \sup_{\boldsymbol{\ell} \in \mathcal{L}^T} \text{Reg}(\mathcal{A}(\boldsymbol{\ell}), \boldsymbol{\ell})$.

One of the fundamental results in online learning is that there exist algorithms $\mathcal{A}$ that guarantee $O(\sqrt{T})$ regret (e.g., online gradient descent), which is the best possible dependency one can hope for in terms of $T$. However, the optimal scaling factor in front of the $\sqrt{T}$ depends on the geometry of the action and loss sets $\mathcal{X}$ and $\mathcal{L}$ and is the primary focus of interest in this paper. To this end, define $\text{Rate}(\mathcal{A}) = \limsup_{T \to \infty} \frac{1}{\sqrt{T}} \cdot \text{Reg}_T(\mathcal{A})$ to be the worst-case scaling factor achieved by the algorithm $\mathcal{A}$, and $\text{Rate}(\mathcal{X}, \mathcal{L}) = \inf_{\mathcal{A}} \text{Rate}(\mathcal{A})$ to be the best possible scaling factor achieved by any algorithm for this action set and loss set. Our goal is to understand how to approximate $\text{Rate}(\mathcal{X}, \mathcal{L})$ and design corresponding optimal algorithms for any choice of action set and loss set.

## 3.2 REGULARIZERS AND FOLLOW-THE-REGULARIZED-LEADER

One of the most popular classes of learning algorithms for online linear optimization is the class of follow-the-regularized-leader algorithms. *Follow-The-Regularized-Leader (FTRL)* is an algorithm parameterized by a convex function $f : \mathcal{X} \to \mathbb{R}$ (the "regularizer") and a learning rate $\eta > 0$ (which we will generally set equal to $1/\sqrt{T}$). At round $t$, it plays the action $x_t$ given by

$$x_t = \arg\min_{x \in \mathcal{X}} \left( \eta f(x) + \sum_{s=1}^{t-1} \langle x, \ell_s \rangle \right). \tag{1}$$

Intuitively, FTRL always plays an action that is approximately the best response to the current empirical loss (with the regularizer preventing this action from overfitting too rapidly to the actions of the adversary). The class of FTRL algorithms contains many popular algorithms for special cases of online linear optimization, including online gradient descent and multiplicative weights.

It can be shown that as long as $f$ is strongly convex, FTRL will incur $O(\sqrt{T})$ regret and thus have non-infinite rate – however, the value of $\text{Rate}(\mathcal{X}, \mathcal{L})$ can depend significantly on the choice of $f$. For example, when $\mathcal{X} = \Delta_d$ and $\mathcal{L} = [0, 1]^d$ (the classic setting for *learning from experts*), it is known that:

- If we use the quadratic regularizer $f(x) = \|x\|^2$, the resulting rate of the FTRL algorithm is $\text{Rate}(\mathcal{A}) = \Theta(\sqrt{d})$. (This corresponds to running online gradient descent).

- If we use the negative entropy regularizer $f(x) = \sum_i x_i \log x_i$, the resulting rate of the FTRL algorithm is $\mathrm{Rate}(\mathcal{A}) = \Theta(\sqrt{\log d})$. (This corresponds to running multiplicative weights / Hedge).

We will soon see that the optimal rate is achieved by some instantiation of FTRL (Theorem 12), and therefore much of our focus will be on computing a suitable regularizer $f$ for a given pair of action set and loss set $(\mathcal{X}, \mathcal{L})$. To this end, it is useful to understand the guarantees the standard analysis of FTRL grants us for a specific choice of regularizer. Before we can state these, we will need to introduce some terminology regarding convex sets and their associated norms.

First, we will make the standard assumption in convex optimization that all of our convex sets are bounded and contain an open ball. In particular, we have the following assumption:

**Assumption 1.** *We assume the action and loss sets are symmetric, they include a ball of radius $r$ and are included in a ball of radius $R$: $B(0, r) \subseteq \mathcal{X}, \mathcal{L} \subseteq B(0, R)$.*

The symmetry assumption allows us to define norms corresponding to $\mathcal{X}$ and $\mathcal{L}$. In general, the norm provided by a bounded symmetric convex set $\mathcal{C}$ is defined as follows:

**Definition 1.** *Given a bounded symmetric convex subset $\mathcal{C} \subseteq \mathbb{R}^d$, we define the natural norm $\|.\|_{\mathcal{C}}$ corresponding to $\mathcal{C}$ as*

$$\forall v \in \mathbb{R}^d, \|v\|_{\mathcal{C}} \triangleq \inf\{\alpha > 0, \tfrac{v}{\alpha} \in \mathcal{C}\}. \tag{2}$$

*It is easy to check that $\|.\|_{\mathcal{C}}$ defined in Equation equation 2 is a norm Leonard & Lewis (2015).*

Given a symmetric convex set $\mathcal{C}$, we can also define a norm on linear functionals over $\mathcal{C}$ by constructing the appropriate dual convex set.

**Definition 2.** *Given a symmetric convex set $\mathcal{C} \subseteq \mathbb{R}^d$, the dual set $\mathcal{C}^c$ is defined as $\mathcal{C}^c \triangleq \{x \in \mathbb{R}^d : \forall y \in \mathcal{C}, \langle x, y \rangle \leq 1\}$.*

*Note that if $\mathcal{C}$ is symmetric, bounded, and full-dimensional, the dual set $\mathcal{C}^c$ is symmetric, bounded, and full-dimensional. The dual norm $\|v\|_{\mathcal{C}^c}$ is the norm corresponding to the dual set.*

We also need to define the notion of strong convexity with respect to an arbitrary norm $\|.\|_{\mathcal{C}}$:

**Definition 3.** *A convex function $f : \mathcal{X} \to \mathbb{R}$ is strongly-convex with respect to norm $\|.\|_{\mathcal{C}}$ if for every $x, y \in \mathcal{X}$ and every sub-gradient $g$ of $f$ at $x$: $f(y) \geq f(x) + \langle y - x, g \rangle + \frac{\alpha}{2} \|y - x\|_{\mathcal{C}}^2$.*

Now we are ready to state the standard regret bound for FTRL with regularizer $f$. As we see, the regret bound depends on both the strong convexity of $f$ with respect to the dual norm of $\mathcal{L}$, and the range of $f$ over $\mathcal{X}$:

**Fact 1.** *[Theorem 5.2 in Hazan et al. (2016)] Let $\mathrm{FTRL}(f)$ be the FTRL algorithm initialized with regularizer $f$ and learning rate $\eta = 1/\sqrt{T}$. If $0 \leq f(x) \leq C^2$ for all $x \in \mathcal{X}$ and $f$ is $\alpha$-strongly-convex with respect to $\mathcal{L}^c$ on $\mathcal{X}$ (see Definition 3), then $\mathrm{Reg}(\mathrm{FTRL}(f)) \leq O(C\sqrt{\alpha T})$.*

### 3.3 Convex Optimization and Oracles

We will in general assume that we have *oracle access* (i.e., access to membership oracles, separation oracles, linear optimization oracles) to the sets $\mathcal{X}$ and $\mathcal{L}$. For a more comprehensive definition of these oracles, see Appendix C.

## 4 Main Result and Overview

Our main contribution is to propose an algorithm for computing a regularizer $g$ such that running FTRL with $g$ achieves the optimal regret of $O\left(\mathrm{Rate}(\mathcal{X}, \mathcal{L})\sqrt{T}\right)$ for the online linear optimization problem, as defined in Section 3.1. In particular, we state our main result in the following theorem.

**Theorem 1** (Algorithmic optimal online linear optimization). *Given access to a linear optimization oracle for $\mathcal{L}$, which can minimize any linear function $c^\top x$ over $\mathcal{L}$ up to accuracy $\delta_{\mathrm{lin}}$ in time $\mathrm{LINO}_{\mathcal{L}}(\delta_{\mathrm{lin}})$, there is a cutting-plane algorithm that runs in time $\left(\frac{dR}{r}\right)^{O(d^2)} \cdot \mathrm{LINO}_{\mathcal{L}}\left(\left(\frac{r}{dR}\right)^{\Theta(d)}\right)$ and calculates a regularizer $g$ which satisfies*

1. $\sup_{x \in \mathcal{X}} |g| = O(\text{Rate}(\mathcal{X}, \mathcal{L})^2)$,

2. $g$ is 1-*strongly convex w.r.t* $\|.\|_{\mathcal{L}^c}$.

*Furthermore, given access to a membership oracle to $\mathcal{X}$ and the regularizer $g$ (which can be precomputed and summarized via a $\exp(O(d^2))$-dimensional vector as described in Section 7) there is a cutting-plane algorithm that runs FTRL with regularizer $g$ with running time $O\left(d^2 \ln^{O(1)} (dRT)\right)$ per round and which guarantees regret $O(\text{Rate}(\mathcal{X}, \mathcal{L})\sqrt{T})$.*

The starting point of our proof of the above theorem is to demonstrate the existence of a regularizer that enables FTRL to achieve the optimal minimax regret, up to a constant factor.

**Theorem 2.** *There exists a regularizer $f_0$ so that running FTRL with $f_0$ yields a regret of $\text{Reg}(\mathbf{x}, \boldsymbol{\ell}) \leq O(\text{Rate}(\mathcal{X}, \mathcal{L})\sqrt{T})$.*

We prove Theorem 2 in Appendix A, where we eliminate the additional $\log(T)$ factor from the regret analysis of the regularizer in Srebro et al. (2011), proving that it achieves the optimal regret bound of $O\left(\text{Rate}(\mathcal{X}, \mathcal{L})\sqrt{T}\right)$, up to universal constants. This improvement is made possible by a novel analytic estimate for the norm growth of certain martingales. In particular, we prove in Theorem 7 that the regularizer from Srebro et al. (2011) can be chosen to be 1-strongly convex with respect to $\|.\|_{\mathcal{L}^c}$ while being bounded by $O\left(\text{Rate}(\mathcal{X}, \mathcal{L})^2\right)$ on the domain $\mathcal{X}$. Theorem 2 then follows from Theorem 7 and Fact 1.

This allows us to restrict our attention to the problem of finding the optimal regularizer over $\mathcal{X}$ which is 1-strongly-convex with respect to $\|.\|_{\mathcal{L}^c}$. To effectively do this optimization, it is important that the resulting regularizer has not only bounded values, but also bounded *gradients*. Note that this is not a priori achieved by the regularizers guaranteed to exist by Theorem 7, and in fact several optimal regularizers used in practice (e.g. the negative entropy regularizer) do have unbounded gradients. Nonetheless, in Section 5 and Appendix B, we demonstrate how to use Gaussian smoothing to obtain a new regularizer that (1) achieves the same optimal regret when used in FTRL, and (2) has smooth derivatives (Theorem 3).

Our next step is to show that we can effectively optimize over the space of smooth convex functions defined over $\mathcal{X}$. To do so, we show that given a near-optimal smooth regularizer $f$, we can approximate it using "quasi-quadratic" functions such that the resulting regularizer $\tilde{f}$ remains (1) $\alpha/2$ strongly convex with respect to $\|.\|_{\mathcal{L}^c}$, and (2) bounded by $O\left(\text{Rate}(\mathcal{X}, \mathcal{L})^2\right)$ on $\mathcal{X}$. Notably, the set of quasi-quadratic functions (with a discretized set of centers) is finite-dimensional, and so the optimal regularizer can be encoded by a finite-dimensional vector $\tilde{\mathcal{I}}$. We carry this out in Section 6.

Finally, in Section D, we demonstrate how to optimize over this set by writing an explicit convex program such that $\tilde{f}$ is a feasible solution to this program, but also such that any feasible solution so that any feasible solution $\mathcal{I}$ from this set yields a regularizer $g^{(\mathcal{I})}$ with near optimal regret. Solving this convex program can be done via standard cutting-plane methods, except for one of the constraints that involves checking whether a candidate regularizer $g$ is $\alpha$-strongly-convex with respect to $\|.\|_{\mathcal{L}^c}$. In Section E, we demonstrate how to construct a separation oracle for this constraint, and finally establish the existence of this algorithm.

As seen in Theorem 1, computing and storing this optimal regularizer takes time that is exponential in the dimension of the problem. In Section 8, we establish a lower bound based on the result of Bhattiprolu et al. (2021) that even checking the strong convexity of the Euclidean norm squared regularizer with respect to $\|.\|_{\mathcal{L}^c}$ requires an exponential number of queries in the dimension.

## 5 A SMOOTH OPTIMAL REGULARIZER

While Theorem 2 promises the existence of an ideal regularizer which achieves the optimal rate, this regularizer is not computable. To design an algorithm, we aim to approximate $f_0$ with a parametric family of functions. At a high level, we plan to accomplish this by locally approximating the regularizer at a finite set of points with simple parameteric functions. Based on Fact 1, our goal is to construct the approximation so that (1) it preserves the strong convexity of $f_0$, (2) it is bounded by $O(\text{Rate}(\mathcal{X}, \mathcal{L})^2)$ on $\mathcal{X}$, ensuring that the resulting regret matches the bound in Theorem 2.

To preserve the strong convexity, a first order approximation of $f_0$ is insufficient as it flattens the function's curvature. Therefore, we rely on a second order approximation of $f_0$ around a discretized set of points $\mathcal{S} \subseteq \mathcal{X}$. For these approximations to remain close to $f_0$ locally around each $x_i \in \mathcal{S}$, we require $f_0$ to have a Lipschitz-continuous Hessian. However, the regularizer from Srebro et al. (2011) does not necessarily possess smooth derivatives. We side-step this issue by proposing an alternative regularizer that not only achieves the optimal rate of $O(\mathrm{Rate}(\mathcal{X}, \mathcal{L})\sqrt{T})$ but also features smooth derivatives 2. This regularizer can then be approximated by our strategy.

**Theorem 3** (Existence of smooth regularizer). *There exists a regularizer $f$ so that running FTRL with $f$ has regret bound $\mathrm{Reg}(\mathrm{FTRL}(f)) \leq O(\mathrm{Rate}(\mathcal{X}, \mathcal{L})\sqrt{T})$. In addition, the derivatives of $f$ are bounded as $|D^i f(x)[v, \ldots, v]| = O(\mathrm{Rate}(\mathcal{X}, \mathcal{L})^2 \frac{d^{i/4}}{r^i})$.*

*Proof.* The proof follows from combining Theorems 8 and 7 with Fact 1. $\qquad\square$

We construct the smooth regularizer $f$ of Theorem 3 by adding Gaussian noise to $f_0$, and prove that (1) the Gaussian smoothing does not impact performance; running mirror descent with $\tilde{f}$ achieves the same regret bound as running mirror descent with $f$, and (2) the derivatives of $f$ are sufficiently smooth due to the Gaussian smoothing (see Theorem 8.)

# 6 APPROXIMATING THE SMOOTH REGULARIZER

Now that we can restrict our attention to smooth regularizers, we can attempt to approximate them via low-degree polynomial functions. Using the derivative bound for the smooth regularizer $f$ in Theorem 3, it is easy to obtain a Hessian $L$-Lipschitz property for $L = \mathrm{Rate}(\mathcal{X}, \mathcal{L})^2 \frac{d^{3/4}}{r^3}$, defined as:

$$\left\| \nabla^2 f(x_0) - \nabla^2 f(x_1) \right\| \leq L \left\| x_0 - x_1 \right\|, \tag{3}$$

for all $x_0, x_1 \in \mathbb{R}^d$. Using the Hessian smooth property in equation 3, we can show that the quadratic approximation of $f$ around $x_0$ remains valid locally. However, we also need to build an approximation for $f$ with the property that it also achieves almost the same maximum on $\mathcal{X}$ as $f_0$. We impose this condition on our approximations by adding a norm-cubic term to the quadratic approximation of $f$ at $x_0$. Hence, our final approximation of $f$ around $x_0$ takes the following form:

$$f_{x_0}(x) = f(x_0) + \langle \nabla f(x_0), x - x_0 \rangle + \frac{1}{2}(x - x_0)^\top \nabla^2 f(x_0)(x - x_0) - \frac{L}{3}\|x - x_0\|^3. \tag{4}$$

We refer to a function of the form in equation 4 as "quasi-quadratic," centered at $x_0$. The intuition for this approximation is that the norm cubic term adds a decay to the Hessian of the function as we move away from $x_0$; this decay guarantees that $f_{x_0}(x)$ is always a lower bound for $f$, and in particular can be estimated by $f$ from above and below with the margin $L\|x - x_0\|^3$. We show this in Lemma 1. On the other hand, this decay is slow enough so that from the $L$-Hessian smoothness of $f$ we can prove that the Hessian of the approximation remains almost the same as the Hessian of $f$, at least locally around $x_0$; therefore, the strong convexity property can be preserved (see Lemma 3.)

**Lemma 1** (estimating $f$ by the approximator). *We have the following relation between the value of $f$ and $f_{x_0}$:*

$$f_{x_0}(x) + \frac{L}{6}\|x - x_0\|^3 \leq f(x) \leq f_{x_0}(x) + \frac{L}{2}\|x - x_0\|^3.$$

The proof of Lemma 1 is in Section F.1. Finally, we combine these local approximations around a discretization set $\mathcal{S}$ in $\mathcal{X}$ by taking their maximum. In particular, we define a piece-wise quasi-quadratic function $\tilde{f}$ to approximate $f$ as $\tilde{f}(x) = \sup_{i \in [N]} f_{x_i}(x)$. The observation is that while $\tilde{f}$ remains strongly convex and suitably bounded on $\mathcal{X}$, it is also efficiently encoded by $f(x_i)$, $\nabla f(x_i)$, and $\nabla^2 f(x_i)$ at discretized points $\mathcal{S} = \{x_i\}_{i=1}^N$, since each $f_{x_i}(x)$ does not use more than zeroth, first, and second order information of $f$ at $x_i$'s. Therefore, we can narrow our search for suitable regularizers from all convex functions on $\mathbb{R}^d$ to the selection of the value, gradient, and Hessian of a piece-wise quasi-quadratic function at a finite set of points. In fact, in the next section we write a convex program to minimize the maximum value of these piecewise quasi-quadratic regularizers.

# 7 A CONVEX PROGRAM FOR CALCULATING AN IDEAL REGULARIZER

In the previous section, we showed how to approximate $f$ with a set of quasi-quadratic approximators, which only uses the value, gradient, and Hessian information of $f$ at a finite set of points $\mathcal{S} = \{x_i\}_{i=1}^N$. Here, we hope to search in the space of such approximators by defining a convex program whose variables are the function's value, gradient and Hessian at $\mathcal{S}$, denoted by $\{r_{x_i}, v_{x_i}, \Sigma_{x_i}\}_{i=1}^N$. Before rigorously defining the program, we first provide motivation for its definition. In particular, we want the instance $\tilde{\mathcal{I}} = \left(\{\tilde{r}_{x_i}\}_{i=1}^N, \{\tilde{v}_{x_i}\}_{i=1}^N, \{\tilde{\Sigma}_{x_i}\}_{i=1}^N\right)$ where $\tilde{r}_{x_i} \triangleq f(x_i), \tilde{v}_{x_i} \triangleq \nabla f(x_i), \tilde{\Sigma}_{x_i} \triangleq \nabla^2 f(x_i)$, corresponding to the smoothed regularizer $f$ in Theorem 2, to be a feasible point. On the other hand, for any instance $\mathcal{I} = (\mathbf{r}, \mathbf{v}, \mathbf{\Sigma}) = \left(\{r_{x_i}\}_{i=1}^N, \{v_{x_i}\}_{i=1}^N, \{\Sigma_{x_i}\}_{i=1}^N\right)$, we can define a regularizer $g_{x_i}^{(\mathcal{I})}(x)$ as

$$g^{(\mathcal{I})}(x) \triangleq \max_{i \in [N]} g_{x_i}^{(\mathcal{I})}(x), \tag{5}$$

where imitating the approximation that we derived for $f$ in equation 4, $g_{x_i}^{(\mathcal{I})}(x)$ denotes a quasi-quadratic function:

$$g_{x_i}^{(\mathcal{I})}(x) = r_{x_i} + \langle v_{x_i}, x - x_i \rangle + \tfrac{1}{2}(x - x_i)^\top \Sigma_{x_i}(x - x_i) - \tfrac{L}{6}\|x - x_i\|^3. \tag{6}$$

With this terminology, it is clear that $\tilde{f} = g^{\tilde{\mathcal{I}}}$. Besides having $\tilde{\mathcal{I}}$ as a feasible point of the program, we also want to impose constraints so that for the optimal solution of the program, $\mathcal{I}^*$, the regularizer $g^{(\mathcal{I}^*)}$ is strongly convex and suitably bounded on $\mathcal{X}$. First, note that from Lemma 9, $\alpha$-strong convexity of $f$ with respect to $\|.\|_{\mathcal{L}^c}$ is equivalent to the condition

$$v^\top \nabla^2 f(x) v \geq \alpha \tag{7}$$

for all $x \in \mathcal{X}$ and $v \in \mathcal{L}$. Hence, we also add the condition $v^\top \Sigma_{x_i} v \geq \alpha, \forall v \in \mathcal{L}$ to the program. While this condition asserts strong convexity of $g^{(\mathcal{I})}$ for all feasible instances $\mathcal{I}$ at the discretization points, it does not guarantee strong convexity elsewhere. The reason is that the approximator in equation 6 is not strongly convex for points far from $x_i$. Therefore, in order to guarantee strong convexity for $g^{(\mathcal{I})}$ everywhere, we need to make sure that at any point $x \in \mathcal{X}$, the maximum in equation 5 is attained by a function $g_{x_i}^{(\mathcal{I})}$ where $x_i$ is sufficiently close to $x$. Building on this observation, we introduce the concept of "locality" for an arbitrary instance $\mathcal{I}$:

**Definition 4.** *We define an instance $\mathcal{I} = (\mathbf{r}, \mathbf{v}, \mathbf{\Sigma})$ as $\epsilon$-local if, for every $x$, $\left\|x_{\hat{i}(x)} - x\right\| = O(\epsilon)$ where $\hat{i}(x) \triangleq \arg\max_{i \in [N]} g_{x_i}^{(\mathcal{I})}(x)$.*

Note that $\epsilon$-locality is guaranteed for $\tilde{f} = g^{(\tilde{\mathcal{I}})}$ by Lemma 1. Specifically, if there is a point $x_i \in \mathcal{S}$ such that $\|x_i - x\| = O(\epsilon)$, then according to Lemma 1, the point $x_{\hat{i}(x)}$ where $g_{x_{\hat{i}(x)}}$ attains its maximum in equation 5 at $x$, must also be within a distance of $O(\epsilon)$ from $x$. To ensure that the maximum equation 5 is attained at an $x_{\hat{i}(x)}$ that is close to $x$, we enforce a slightly relaxed version of the lower bound from Lemma 1 on $g^{(\mathcal{I})}$ at the discretization points:

$$g_{x_i}^{(\mathcal{I})}(x_j) + \tfrac{15L}{96}\|x_j - x_i\|^3 \leq r_{x_j}, i, j = 1, \ldots, N. \tag{8}$$

As noted in Lemma 1, $\tilde{f}$ satisfies the inequality $f_{x_0}(x) + \tfrac{L}{6}|x - x_0|^3 \leq f(x)$. The reason we apply a slightly weaker version of this inequality in equation 8 will become evident when we design a separation oracle for the feasibility set of the convex program. At a high level, this condition ensures that not only is $\tilde{\mathcal{I}}$ a feasible instance for our program, but that a small neighborhood around it also remains feasible. As we will see, even after enforcing the condition in equation 8, an arbitrary feasible instance $\mathcal{I}$ does not achieve $O(\epsilon)$-locality like $\tilde{\mathcal{I}}$. Instead, we can only prove that it is $O(\epsilon^{1/3})$-local (see Lemma 2). The reason is that equation 8 is only enforced at the discretization points, whereas $\tilde{f}$ satisfies it for any $x \in \mathcal{X}$ as shown in Lemma 1.

Finally, we aim to minimize the maximum value of $g^{(\mathcal{I})}$ over $\mathcal{X}$ to obtain a suitable regularizer for FTRL. As mentioned earlier, we smooth the theoretical regularizer $f_0$ from Srebro et al. (2011) by

adding Gaussian noise, resulting in $f$, which ensures bounded gradients and Hessians. To achieve a similar smoothness condition on the regularizer $g^{(\mathcal{I})}$ that correspond to a feasible instance of our program, we enforce the conditions $\|v_{x_i}\|_\infty \leq c_0$ and $\Sigma_{x_i} \preccurlyeq c_2 I$ for constants $c_0, c_2$ (we use the infinity norm instead of the 2-norm to maintain a linear constraint.) With the discretization set $\mathcal{S} = \{x_i\}_{i=1}^N$ fixed, the final program is structured as follows:

$$\text{minimize } r \tag{9}$$

$$
\begin{aligned}
\text{subject to } & r_{x_i} + \langle v_{x_i}, x_j - x_i \rangle + \tfrac{1}{2}(x_j - x_i)^\top \Sigma_{x_i}(x_j - x_i) - \tfrac{17L}{96}\|x_j - x_i\|^3 \leq r_{x_j} && \forall i,j \in [N] \\
& \|v_{x_i}\|_\infty \leq c_0 && i \in [N] \\
& \Sigma_{x_i} \preccurlyeq c_2 I && \forall i \in [N] \\
& v^\top \Sigma_{x_i} v \geq \alpha && \forall v \in \mathcal{L}, \ \forall i \in [N] \\
& r \geq r_{x_i} && \forall i \in [N] \\
& r, r_{x_i} \leq C_0 && \forall i \in [N].
\end{aligned}
$$

Next, to establish the locality property for feasible points of the program, we state in Lemma 2 that for any arbitrary $x \in \mathcal{X}$, the maximum in equation 5 is attained at a discretization point $x_i \in \mathcal{S}$ that is not too far from $x$. Specifically, given that every point in $\mathcal{X}$ has a discretization point $x_i$ within a distance of $\epsilon$, we show that the maximum in equation 5 is achieved by $x_{\hat{i}}$ which is no further than $O(\epsilon^{1/3})$ from $x$. Additionally, we prove that the value of $g^{(\mathcal{I})}$ at $x$ is close to $g^{(\mathcal{I})}_{x_i}(x)$.

**Lemma 2** (Convex program feasibility $\to$ Locality of regularizer $g$). *Assume that $\mathcal{I} = (\mathbf{r}, \mathbf{v}, \mathbf{\Sigma})$ is feasible for LP equation 9, for $\epsilon$ satisfying $\epsilon \leq \gamma_2 \left\{ \frac{L}{\sqrt{d}c_0}, \frac{L}{c_0\sqrt{d}c_2^3}, \frac{L}{c_2}, \sqrt{c_0\sqrt{d}}, \frac{c_0\sqrt{d}}{c_2} \right\}$, then suppose for $x_i, x_j$ and $x \in \mathcal{X}$ we have $\|x_i - x\| \leq \epsilon$ and $\|x_j - x\| \geq \gamma \left( \frac{\epsilon\sqrt{d}c_0}{L} \right)^{1/3}$ for some universal constant $\gamma$, then*

$$g^{(\mathcal{I})}_{x_i}(x) > g^{(\mathcal{I})}_{x_j}(x) + \sqrt{d}c_0\epsilon,$$

*and if $\|x_j - x\| \leq \gamma \left( \frac{\epsilon\sqrt{d}c_0}{L} \right)^{1/3}$, then*

$$|g^{(\mathcal{I})}_{x_j}(x_i) - g^{(\mathcal{I})}_{x_j}(x)| \leq \gamma_2\sqrt{d}c_0\epsilon,$$

*for some constant $\gamma_2$.*

The proof can be found in Section F.2. To prove strong convexity of $g^{(\mathcal{I})}$ for a feasible point $\mathcal{I}$, we must first establish the strong convexity of the local approximators $g^{(\mathcal{I})}_{x_i}$, defined in equation 6. This is demonstrated in Lemma 3 below. Specifically, we prove that if the quadratic form of the Hessian variable $\Sigma_{x_i}$ is lower bounded by the norm squared $\|.\|^2_{\mathcal{L}^c}$ in all directions, then $g^{(\mathcal{I})}_{x_i}(x)$ is strongly convex locally around $x_i$.

**Lemma 3.** *[Local strong convexity of the approximators] Suppose the PSD matrix $\Sigma$ is such that for all $v$, $v^\top \Sigma v \geq \alpha \|v\|^2_{\mathcal{L}^c}$. Then, the function*

$$g(x) = r + \langle v, x - x_0 \rangle + \tfrac{1}{2}(x - x_0)^\top \Sigma (x - x_0) - \tfrac{L}{6}\|x - x_0\|^3$$

*for arbitrary $x_0, v, r, L$ is $\alpha/2$-strongly convex with respect to $\|.\|_{\mathcal{L}^c}$ in the neighborhood $\|x - x_0\| \leq \frac{\alpha}{2R^2L}$. Consequently, if $f$ is $\alpha$-strongly convex with respect to $\|.\|_{\mathcal{L}}$, then $f_{x_0}(x)$ is $\frac{\alpha}{2}$ strongly convex with respect to $\|.\|_{\mathcal{L}}$ for $\|x - x_0\| \leq \frac{\alpha}{2R^2L}$.*

The proof of Lemma 3 is in Section F.3. Finally, by combining Lemmas 14 and 2, we show that the barrier $g^{(\mathcal{I})}$ constructed from a feasible point of the matrix program has a suitable upper bound on $\mathcal{X}$, satisfying the desired strong convexity. Additionally, we prove that the feasible region can be approximated both from the inside and outside by Euclidean balls, a key property necessary for constructing a separation oracle for the feasible set later.

**Theorem 4** (Convex program solution $\rightarrow$ optimal regularizer). *Assume we are given a smooth barrier function $f : \mathbb{R}^d \rightarrow \mathbb{R}$ with $|f(x)| \leq C^2, \forall x \in \mathcal{X}$, which is $\tilde{c}_1$ Lipschitz, $\tilde{c}_2$ gradient Lipschitz, $\tilde{L}$ Hessian Lipschitz, and $\alpha \|.\|_{\mathcal{L}^c}$-strongly convex in $\mathcal{X}$. Additionally, if for every two points in the cover $x_i, x_j \in \tilde{\mathcal{X}}$ we have $\|x_i - x_j\| \geq \bar{\epsilon}$, then the convex program in equation 9 with $c_0 = \tilde{c}_1 + L\bar{\epsilon}^3, c_2 = \tilde{c}_2 + L\bar{\epsilon}^3, L = \tilde{L}, C_0 = C^2 + L\bar{\epsilon}^3$, and discretization parameter $\epsilon \leq \gamma_3 \min \left\{ \frac{L}{\sqrt{d}c_1}, \frac{L}{c_1\sqrt{d}c_2{}^3}, \frac{L}{c_2}, \sqrt{c_1\sqrt{d}}, \frac{c_1\sqrt{d}}{c_2}, \frac{\alpha^3}{512R^6L^2c_1\sqrt{d}} \right\}$ for small enough constant $\gamma_3$ is feasible. Furthermore, the function $g^{(\mathcal{I}^*)}$, corresponding to the optimal solution $\mathcal{I}^* = (\mathbf{r}^*, \mathbf{v}^*, \mathbf{\Sigma}^*)$ is convex and satisfies the following properties:*

1. *$|g^{(\mathcal{I}^*)}(x)| \leq C^2 + \gamma_2\epsilon\sqrt{d}c_0$ for constant $\gamma_2$.*

2. *For any feasible instance $\mathcal{I} \in \mathcal{P}_{\mathcal{I}}$, $g^{(\mathcal{I})}(x)$ is $\frac{\alpha}{2}$ strongly convex with respect to $\|.\|_{\mathcal{L}^c}$.*

3. *$B_{L\bar{\epsilon}^3/288}(\tilde{\mathcal{I}}) \subseteq P_{\mathcal{I}} \subseteq B_{2\sqrt{(N+1)C_0{}^2 + Nd(c_0^2 + c_2^2)}}(\tilde{\mathcal{I}})$.*

Proof of Theorem 4 can be found in Section F.4.

## 8 LOWER BOUND ON MEMBERSHIP ORACLE QUERY COMPLEXITY FOR $\mathcal{L}$

In the above sections we demonstrated an algorithm for computing an optimal regularizer that runs in time $\exp(O(d^2))$. In this final section, we show that this is in some sense necessary, by showing that just checking the $\alpha$-strong convexity of a given regularizer $g$ with respect to $\|.\|_{\mathcal{L}^c}$ at point $x \in \mathcal{X}$ requires an exponential number of queries to a membership oracle $\mathrm{MEM}_{\mathcal{L}}(\delta)$. In particular, even in the simple case where $\nabla^2 g(x) = I$ (i.e., the quadratic regularizer), an exponential number of queries is needed. The lower bound is a reduction to Theorem 1.2 in Bhattiprolu et al. (2021).

**Theorem 5** (Exponential lower bound). *Given $\epsilon$, for large enough dimension $d$, there exists a distribution over convex bodies $\mathcal{L}$ such that for every fixed set of queried points $S \subseteq \mathbb{R}^d$,*

1. *$\mathbb{P}_{\mathcal{L}}\left(S \cap \{v| \; \|v\|_{\mathcal{L}} \leq 1\} = S \cap B_1(0)\right) \geq 1 - \epsilon$*

2. *There exists direction $\tilde{v}$ with $\|\tilde{v}\|_{\mathcal{L}^c} = 1$ such that $\|\tilde{v}\|_2 \leq \frac{1}{d^{1-\epsilon}}$,*

*where $B_1(0)$ is the Euclidean ball with radius 1.*

The proof of Theorem 5 is provided in Section F.5. At a high level, Theorem 5 asserts that there exists a distribution over norm balls $\mathcal{L}$ such that (1) even with $e^{d^{1-\epsilon}}$ queries it is not insufficient to distinguish between $\mathcal{L}$ and the Euclidean unit ball, while (2) the Identity Hessian is not $\alpha = \frac{1}{d^{1-\epsilon}}$ strongly convex with respect to the dual norm $\|.\|_{\mathcal{L}^c}$.

Of course, it is possible that there is a method for computing the optimal regularizer that sidesteps to need to be able to verify how convex an arbitrary function is – we leave this as an interesting open problem.

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

## A  IDEAL REGULARIZER AND PROVING BETTER MARTINGALE TYPE FOR $p = 2$

Here, we state the existence of an ideal regularizer such that running FTRL with this regularizer achieves the optimal rate up to a constant. This result is adapted from Srebro et al. (2011), except that they prove the same regularizer results in a regret bound which is off by a logarithmic factor of $\log(T)$; this log factor is indeed not desirable for our purpose as we are interested in long time horizon regimes when $T$ can potentially be exponentially large in dimension. Our contribution here is that we improve the result of Srebro et al. (2011) for $p = 2$ case and shave off this log factor. We further show a type of continuity condition for this ideal regularizer that we use for our smoothing arguments in Section B.

First, we state the result of Sridharan & Tewari (2010); Rakhlin et al. (2010) that we build upon; it is known from the work of Sridharan & Tewari (2010); Rakhlin et al. (2010) that the optimal rate for adversarial online linear optimization translates into a property on the growth of the norm $\|.\|_{\mathcal{X}^c}$ of an arbitrary Rademacher martingale sequence. We state this property rigorously in Theorem 6, which is stated as Theorem 4 in Srebro et al. (2011).

**Theorem 6** (Restatement of Theorem 4 in Srebro et al. (2011))**.** *Given the optimal rate for online linear optimization with action and loss sets $\mathcal{X}, \mathcal{L} \in \mathbb{R}^d$ is $O(C\sqrt{T})$, then for a Rademacher random vector $\epsilon \in \{\pm\}^n$ and any sequence of functions $x_i(\epsilon) : \{\pm\}^i \to \mathbb{R}^d$, where $x_i$ is a function of the first $i$ coordinates in $\epsilon$, we have*

$$\mathbb{E}\left\|\sum_i \epsilon_i x_i(\epsilon)\right\|_{\mathcal{X}^c} \leq O(C) \sup_{0 \leq i \leq n} \sup_{\epsilon} \|x_i(\epsilon)\|_{\mathcal{L}}. \tag{10}$$

The main contribution of authors in Srebro et al. (2011) is that they translate equation 10 to the existence of a suitable barrier for mirror descent. In particular, they prove the following key Lemmas 4, 7. We start with Lemma 4 which translates property equation 10 to a more refined argument about the growth of martingale norms that are defined based on the action and loss sets.

**Lemma 4** (Restatement of Lemma 12 in Srebro et al. (2011) for $r = 2$)**.** *For $1 < r < 2$, if there exists a constant $C > 0$ such that for any natural number $n$ and any sequence of mappings $(x_i)_{i=1}^n$, $x_i : \{\pm\}^i \to \mathbb{R}^d$ and Rademacher random vector $\epsilon \in \{\pm\}^n$ satisfy*

$$\mathbb{E}\left\|\sum_{i=1}^n \epsilon_i x_i(\epsilon)\right\|_{\mathcal{X}^c} \leq C n^{1/r} \sup_{0 \leq i \leq n} \sup_{\epsilon} \|x_i(\epsilon)\|_{\mathcal{L}},$$

*then for $p < r$ and $\alpha_p = \frac{20C}{r - p}$, for any sequence $(x_i)_{i=1}^n$ as described above, we have the following inequality:*

$$\mathbb{E}\left\|\sum_{i=1}^n \epsilon_i x_i(\epsilon)\right\|_{\mathcal{X}^c} \leq \alpha_p \sup_{\epsilon} \left(\sum_i \|x_i(\epsilon)\|_{\mathcal{L}}^p\right)^{1/p}. \tag{11}$$

The next Lemma states how authors in Srebro et al. (2011) translate the property in Equation equation 11 to the existence of the ideal regularizer:

**Lemma 5** (Restatement of Lemma 11 in Srebro et al. (2011))**.** *For constant $\tilde{C}$, the following statements are equivalent:*

1. *For all $n$ and sequence of mappings $(x_i)_{i=0}^n$ where $x_i : \{\pm\}^{i-1} \to \mathbb{R}^d$:*

$$\mathbb{E}_\epsilon \left\|\sum_{i=1}^n \epsilon_i x_i(\epsilon)\right\|_{\mathcal{X}^c}^p \leq \tilde{C}^p \left(\sum_{i=1}^n \mathbb{E}\|x_n(\epsilon)\|_{\mathcal{L}}^p\right)$$

2. *There exists a 2-homogeneous non-negative convex function $f_0$ on $\mathbb{R}^d$ which is 1-strongly convex w.r.t $\|.\|_{\mathcal{L}^c}$ and $\forall x, \frac{1}{q}\|x\|_{\mathcal{L}^c}^q \leq f_0(x) \leq \frac{\tilde{C}^q}{q}\|x\|_{\mathcal{X}}^q$, where $\frac{1}{p} + \frac{1}{q} = 1$.*

The existence of such regularizer from Lemma 5 then implies a $\tilde{C}T^{1-\frac{1}{p}}$ regret bound for FTRL. Nonetheless, the reason they end up with a $\log(T)$ factor in the regret is that they need to use Lemma 4 with a power $p < 2$ slightly less than two, as the constant $\alpha_p$ reciprocally depends on $2 - p$, so $p$ has to be $\Theta(1/\log(T))$ less than 2. We improve Lemma 4 in Lemma 6 below, for the case of $p = 2$, and shave off the $\alpha_p$ factor which is causing the additional $\log(T)$. This enables us to show a tighter upper bound for the regularizer on domain $\mathcal{X}$ in Theorem 7.

**Lemma 6** (Improving the Martingale Type for $p = 2$). *Suppose for the norm $\|.\|_{\mathcal{X}^c}$ we have*

$$\mathbb{E}\left\|x_0 + \sum_{i=1}^n \epsilon_i x_i(\epsilon)\right\|_{\mathcal{X}^c} \leq D(n+1)^{1/2} \sup_{0 \leq i \leq n} \sup_\epsilon \|x_i(\epsilon)\|_{\mathcal{L}}, \tag{12}$$

*for arbitrary vector valued functions $x_n : \{\pm 1\}^{n-1} \to \mathbb{R}^d$ and Rademacher sequence $(\epsilon_i)_{i=1}^n$, $\epsilon_i \sim \pm 1$. Then, we have*

$$\mathbb{E}\left\|x_0 + \sum_{i=1}^n \epsilon_i x_i(\epsilon)\right\|_{\mathcal{X}^c} \leq D\left(\sum_{i=1}^n \|x_i(\epsilon)\|_{\mathcal{L}}^2\right)^{1/2}.$$

*Proof.* First, note that if we average equation 12 over $x_0$ and $-x_0$ and extend the functions $x_i(\epsilon)$ to also depend on a Rademacher variable $\epsilon_0$ at time zero, then we get

$$\mathbb{E}\left\|\sum_{i=0}^n \epsilon_i x_i(\epsilon)\right\|_{\mathcal{X}^c} \leq D(n+1)^{1/2} \sup_{0 \leq i \leq n} \sup_\epsilon \|x_i(\epsilon)\|_{\mathcal{L}}. \tag{13}$$

Now let $c_i = \|x_i\|_{\mathcal{L}}$. Take a fresh rademacher sequence $(\tilde{\epsilon}_j)_{j=1}^\infty$. We will define the sequence $(\epsilon_i)_{i=1}^n$ based on the randomness of $\tilde{\epsilon}_j$'s: define $\hat{\epsilon}_i = 1$ if $\sum_{j=t_i+1}^{t_{i+1}} \tilde{\epsilon}_j \geq \frac{\|x_i\|}{\delta}$ and $\hat{\epsilon}_i = -1$ if $\sum_{j=t_i+1}^{t_{i+1}} \tilde{\epsilon}_j \leq -\frac{\|x_i\|}{\delta}$. From symmetry, it is easy to check that $\epsilon_i$'s are indeed i.i.d distributed uniformly on $\{\pm 1\}$. Next, for a given positive $\delta > 0$, define the sequence of indices $(t_i)_{i=1}^n$ and the alternative sequence $(\tilde{x}_i)_{i=0}^m$ such that for all $i$, $\tilde{x}_{t_i} = \tilde{x}_{t_i+1} = \cdots = \tilde{x}_{t_{i+1}-1} = \frac{x_i}{\|x_i\|_{\mathcal{L}}}\delta$, and $t_i$ is the first index such that $|\sum_{j=t_i+1}^{t_{i+1}} \tilde{\epsilon}_j| \geq \frac{\|x_i\|}{\delta}$. Now from this definition. we have that $\tilde{x}_i$'s satisfy

$$\left\|x_i - \sum_{j=t_i+1}^{t_{i+1}} \tilde{x}_j\right\|_{\mathcal{X}^c} \leq \delta \left\|\frac{x_i}{\|x_i\|_{\mathcal{L}}}\right\|_{\mathcal{X}^c}. \tag{14}$$

But for $t_{\text{sum}} = \sum_{i=0}^n t_i$ equation 14 implies:

$$\left\|\sum_{i=0}^n \epsilon_i x_i(\epsilon) - \sum_{j=0}^{t_{\text{sum}}} \tilde{\epsilon}_j \tilde{x}_j\right\|_{\mathcal{X}^c} \leq (n+1)\delta \max_{i=0}^n \left\|\frac{x_i}{\|x_i\|_{\mathcal{L}}}\right\|_{\mathcal{X}^c}.$$

The key observation is for all $i \in [n]$, the distribution of $t_i$ is sub-exponential and the sum concentrates around its expectation. In particular,

$$\mathbb{P}\left(t_i \geq \kappa \left(\frac{\|x_i\|_{\mathcal{L}}}{\delta}\right)^2\right) \leq e^{-O(\kappa)}. \tag{15}$$

It is sufficient for us to show that the sum $\sum_{i=1}^n t_i$ is at most $O\left(\sum_{i=1}^n \left(\frac{\|x_i\|_{\mathcal{L}}}{\delta}\right)^2\right)$ with at least constant probability $p$. Call this event $\mathcal{E}$. First, we use Chebyshev inequaility to show $\mathbb{P}(\mathcal{E}) = \Omega(1)$. Note that Equation equation 15 imlies

$$\mathbb{E}t_i^2 = O\left(\frac{\|x_i\|_{\mathcal{L}}}{\delta}\right)^4,$$

which implies

$$Var(\sum_i t_i) = O\left(\sum_i \left(\frac{\|x_i\|_{\mathcal{L}}}{\delta}\right)^4\right).$$

Therefore, from Chebyshev inequality

$$\mathbb{P}\left(\sum_{i=1}^{n} t_i \geq \sum_{i=1}^{n} \left(\frac{\|x_i\|_{\mathcal{L}}}{\delta}\right)^2 + l\sqrt{\sum_{i=1}^{n} \left(\frac{\|x_i\|_{\mathcal{L}}}{\delta}\right)^4}\right) \leq \frac{1}{l^2},$$

which implies

$$\mathbb{P}\left(\sum_{i=1}^{n} t_i \geq (l+1)\sum_{i=1}^{n} \left(\frac{\|x_i\|_{\mathcal{L}}}{\delta}\right)^2\right) \leq \frac{1}{l^2},$$

hence we showed that $\mathcal{E}$ happens with at least constant probability. Furthermore, It is easy to check that conditioned on $\mathcal{E}$, $\epsilon_i$'s are still Rademacher variables. On the other hand, using equation 13 for sequence $(\tilde{x}_i)$ and $m = \Theta\left(\sum_{i=0}^{n} \left(\frac{\|x_i\|_{\mathcal{L}}}{\delta}\right)^2\right)$:

$$\mathbb{E}\left\|\sum_{j=1}^{m} \tilde{\epsilon}_j \tilde{x}_j(\tilde{\epsilon})\right\|_{\mathcal{X}^c} \leq Dm^{1/2} \sup_{0 \leq i \leq n} \sup_{\epsilon} \|\tilde{x}_i(\epsilon)\|_{\mathcal{L}}. \tag{16}$$

but from positivity of norm

$$\mathbb{E}\left\|\sum_{j=1}^{m} \tilde{\epsilon}_j \tilde{x}_j(\tilde{\epsilon})\right\|_{\mathcal{X}^c} \geq \mathbb{E}\left[\left\|\sum_{j=1}^{m} \tilde{\epsilon}_j \tilde{x}_j(\tilde{\epsilon})\right\|_{\mathcal{X}^c} \bigg| \mathcal{E}\right] \mathbb{P}(\mathcal{E})$$

$$\geq \mathbb{E}\left[\left\|\sum_{i=1}^{n} \epsilon_i x_i(\epsilon)\right\|_{\mathcal{X}^c} \bigg| \mathcal{E}\right] \mathbb{P}(\mathcal{E}) - (n+1)\delta \max_{i=0}^{n} \left\|\frac{x_i}{\|x_i\|_{\mathcal{L}}}\right\|_{\mathcal{X}^c}$$

$$= \mathbb{E}\left\|\sum_{i=1}^{n} \epsilon_i x_i(\epsilon)\right\|_{\mathcal{X}^c} \mathbb{P}(\mathcal{E}) - (n+1)\delta \max_{i=0}^{n} \left\|\frac{x_i}{\|x_i\|_{\mathcal{L}}}\right\|_{\mathcal{X}^c}$$

$$\geq \frac{1}{2}\mathbb{E}\left\|\sum_{i=1}^{n} \epsilon_i x_i(\epsilon)\right\|_{\mathcal{X}^c} - (n+1)\delta \max_{i=0}^{n} \left\|\frac{x_i}{\|x_i\|_{\mathcal{L}}}\right\|_{\mathcal{X}^c}.$$

Note that the equality in the third line above is because the size of $t_i$'s is independent of $\epsilon$'s. Plugging this back into equation 16

$$\mathbb{E}\left\|\sum_{i=1}^{n} \epsilon_i x_i(\epsilon)\right\|_{\mathcal{X}^c} \leq \Theta\left(D\sqrt{\sum_{i=0}^{n} \|x_i\|_{\mathcal{L}}^2}\right) + (n+1)\delta \max_{i=0}^{n} \left\|\frac{x_i}{\|x_i\|_{\mathcal{L}}}\right\|_{\mathcal{X}^c}.$$

Sending $\delta \to 0$ finishes the proof. $\qquad\qquad\square$

Next we state and prove Lemma 7. This Lemma in similar to Lemma 7 for the case $p = 2$, i.e. it translates the margtingale property to the existence of an ideal regularizer, except that we show an additional useful Lipschitz property for the regularizer which we use for smoothing the regularizer in Section B. The proof of Theorem 7 directly follows from combining Lemmas 7 and 6.

**Lemma 7** (Martingale type $\to$ ideal regularizer). *For constant $C$, the following statements are equivalent:*

1. *For all $n$ and sequence of mappings $(x_i)_{i=0}^{n}$ where $x_i : \{\pm\}^{i-1} \to \mathbb{R}^d$:*

$$\mathbb{E}_{\epsilon}\left\|x_0 + \sum_{i=1}^{n} \epsilon_i x_i(\epsilon)\right\|_{\mathcal{X}^c}^2 \leq C^2\left(\|x_0\|_{\mathcal{L}}^2 + \sum_{i=1}^{n} \mathbb{E}\|x_n(\epsilon)\|_{\mathcal{L}}^2\right)$$

2. *There exists a $2$-homogeneous non-negative convex function $f$ on $\mathbb{R}^d$ which is $\alpha$-strongly convex w.r.t $\|.\|_{\mathcal{L}^c}$ and $\forall x, \frac{1}{2}\|x\|_{\mathcal{L}^c}^2 \leq f_0(x) \leq \frac{C^2}{2}\|x\|_{\mathcal{X}}^2$. Furthermore, $f$ is Lipschitz continuous as*

$$|f_0(x_1) - f_0(x_2)| \leq C^2 \|x_1 - x_2\|_{\mathcal{X}} \left(\|x_1\|_{\mathcal{X}} \vee \|x_2\|_{\mathcal{X}}\right).$$

*Proof.* This is Lemma 11 in Srebro et al. (2011), except that we are claiming an additional Lipschitz continuity here for $f_0$, which we need to show regularity properties for the gaussian smoothed function later on. To show the Lipschitz continuity, we note that from the proof of Lemma 11 in Srebro et al. (2011), $f_0$ is defined as the Fenchel dual of a barrier $f_0^*$, i.e. $f_0(x) = \sup\langle x, z\rangle - f_0^*(z)$, where $\frac{1}{C^2}\|x\|_{\mathcal{X}^c}^2 \leq f_0^*(x) \leq \|x\|_{\mathcal{L}}^2$. Therefore, defining $z(x) \triangleq \arg\max_z \langle x, z\rangle - f_0^*(z)$, we have

$$0 \leq f_0(z(x)) \leq \|x\|_{\mathcal{X}} \|z(x)\|_{\mathcal{X}^c} - \frac{1}{C^2}\|z(x)\|_{\mathcal{X}^c}^2,$$

which implies

$$C^2 \|x\|_{\mathcal{X}} \geq \|z(x)\|_{\mathcal{X}^c}.$$

Therefore, for $x_1, x_2 \in \mathcal{X}$ we have

$$f_0(x_1) \geq \langle x_1, z(x_2)\rangle - f_0^*(z(x_2)) \geq \langle x_2, z(x_2)\rangle - f_0^*(z(x_2)) - \|x_1 - x_2\|_{\mathcal{X}} \|z(x_2)\|_{\mathcal{X}^c}$$
$$\geq f_0(x_2) - C^2 \|x_1 - x_2\|_{\mathcal{X}} \|x_2\|_{\mathcal{X}}.$$

Noting the reverse symmetric inequality $f_0(x_2) \geq f_0(x_1) - C^2 \|x_1 - x_2\|_{\mathcal{X}} \|x_1\|_{\mathcal{X}}$ completes the proof. $\square$

## B  SMOOTHING THE REGULARIZER

The goal of this section is to show the existence of a regularizer which enables FTRL to achieve the optimal regret for arbitrary pair $(\mathcal{X}, \mathcal{L})$ of action and loss sets which also has smooth derivatives. We achieve this by using Gaussian smoothing of the regularizer $f_0$ from Srebro et al. (2011). First, we state Theorem in which we prove that FTRL with this regularizer indeed achieves the optimal rate $O\left(\text{Rate}(\mathcal{X}, \mathcal{L})\sqrt{T}\right)$; note that this is a $\log(T)$ improvement over the result of Srebro et al. (2011), and in addition the regularizer satisfies a desirable Lipschitz property. We then proceed to smooth this regularizer by adding Gaussian noise and showing the smoothness properties we want.

**Theorem 7** (Existence of an ideal regularizer for mirror descent). *There exists a 2-homogeneous continuous regularizer $f_0 : \mathbb{R}^d \to \mathbb{R}$ which satisfies*

*1. $\max_{x\in\mathcal{X}} |f_0(x)| \leq O(\text{Rate}(\mathcal{X}, \mathcal{L})^2)$*

*2. $f_0$ is 1-strongly convex w.r.t $\|.\|_{\mathcal{L}^c}$ on $\mathcal{X}$, where $\|.\|_{\mathcal{L}^c}$ is the dual norm of $\|.\|_{\mathcal{L}}$.*

*3. $f_0$ satisfies the following Lipschitz continuity condition: $\forall x_1, x_2$:*

$$|f_0(x_1) - f_0(x_2)| \leq O(\text{Rate}(\mathcal{X}, \mathcal{L})^2) \|x_1 - x_2\|_{\mathcal{X}} (\|x_1\|_{\mathcal{X}} \vee \|x_2\|_{\mathcal{X}}).$$

*Proof.* Directly from the relation between optimal rate of online optimization and Equation equation 10, which we state in Theorem 6, with Lemmas 7 and 6. $\square$

For the regularizer $f_0$ given by Theorem 7, we define the Gaussian smoothed function $f : \mathbb{R}^d \to \mathbb{R}$:

$$f(x) = \mathbb{E}_{y\sim N(x,\sigma^2 I)} f_0(y). \tag{17}$$

We start by showing that strong convexity property with respect to arbitrary norms is inherited for $f_0$ to $f$.

**Lemma 8** (Strong convexity of the smoothed function). *If $f_0$ is $\alpha$ strongly convex w.r.t $\|.\|_{\mathcal{L}^c}$, the $f$ is also $\alpha$ strong convex w.r.t $\|.\|_{\mathcal{L}^c}$.*

*Proof.* From $\alpha$ strong convexity of $f$, for $0 \leq \gamma \leq 1$ we have

$$f_0(\gamma x_1 + (1-\gamma)x_2) \leq \gamma f_0(x_1) + (1-\gamma)f_0(x_2) - \alpha \frac{\gamma(1-\gamma)}{2}\|x_1 - x_2\|^2.$$

Now consider the gaussian random variable $\eta \sim N(0, \sigma^2 I)$ and write $\tilde{f}(x_1) = \mathbb{E}_\eta f(x_1 + \eta)$, $\tilde{f}(x_2) = \mathbb{E}_\eta f(x_2 + \eta)$. Then

$$f(\gamma x_1 + (1-\gamma)x_2) = \mathbb{E}_\eta f_0(\gamma x_1 + (1-\gamma)x_2 + \eta)$$
$$= \mathbb{E} f_0(\gamma(x_1 + \eta) + (1-\gamma)(x_2 + \eta))$$
$$\leq \gamma \mathbb{E} f_0(x_1 + \eta) + (1-\gamma)\mathbb{E} f_0(x_2 + \eta) - \alpha \frac{\gamma(1-\gamma)}{2} \|x_1 - x_2\|_{\mathcal{L}^c}^2$$
$$= \gamma f_0(x_1) + (1-\gamma)f_0(x_2) - \alpha \frac{\gamma(1-\gamma)}{2} \|x_1 - x_2\|_{\mathcal{L}^c}^2 .$$

$\square$

**Lemma 9** (Strong convexity $\rightarrow$ Hessian lower bound). *If $f$ is twice continuously differentiable and $\alpha$ strongly convex with respect to $\|.\|_{\mathcal{L}^c}$, then for its hessian at arbitrary point $x$ and arbitrary direction $v$ we have*

$$v^\top \nabla^2 f(x) v \geq \|v\|_{\mathcal{L}^c}^2 . \tag{18}$$

*Proof.* From Taylor series around $x_1$ at points $x_2$ and $\gamma x_1 + (1-\gamma)x_2$:

$$f(x_2) = f(x_1) + \langle \nabla f(x_1), x_2 - x_1 \rangle + \frac{1}{2}(x_2 - x_1)^\top \nabla^2 f(x_1)(x_2 - x_1) + o(\|x_2 - x_1\|^2),$$

$$f(\gamma x_1 + (1-\gamma)x_2)$$
$$= f(x_1) + \langle \nabla f(x_1), (1-\gamma)(x_2 - x_1) \rangle + \frac{1}{2}(1-\gamma)^2 (x_2 - x_1)^\top \nabla^2 f(x_1)(x_2 - x_1) + o(\|x_2 - x_1\|^2).$$

Therefore

$$\gamma f(x_2) + (1-\gamma)f(x_1) - f(\gamma x_1 + (1-\gamma)x_2) = \frac{1}{2}\gamma(1-\gamma)(x_2 - x_1)^\top \nabla^2 f(x_1)(x_2 - x_1) + o(\|x_2 - x_1\|^2).$$

Therefore, $\alpha$ strong convexity is equivalent to equation 18 for all directions $v$. $\square$

**Lemma 10** (Norm and norm squared Gaussian integral). *Given a two-homogeneous function $f_0$ satisfying 1 and $\max_{x \in \mathcal{X}} |f_0(x)| \leq C^2$, then for $f$ defined in equation 17*

$$|f(x)| \leq \frac{C^2}{r^2}\sigma^2 d + C^2 \|x\|_{\mathcal{X}}^2 ,$$

$$\mathbb{E}_{y \sim N(x, \sigma^2 I)} f_0(y)^2 \leq 8C^4 \left( \|x\|_{\mathcal{X}}^4 + \frac{4}{r^4} d\sigma^4 \right).$$

*Proof.* Note that from the property (1) in Theorem 7 and the 2-homogeneity of $f_0$, we have for all $y \in \mathbb{R}^d$, $f_0(y) \leq C^2 \|y\|_{\mathcal{X}}^2$. Now using triangle inequality and Lemma 11, we can write

$$|f(x)| \leq \mathbb{E}_{y \sim N(x, \sigma^2 I)} |f_0(y)|$$
$$\leq \mathbb{E} C^2 \|y\|_{\mathcal{X}}^2$$
$$\leq \mathbb{E} C^2 \|y - x\|_{\mathcal{X}}^2 + C^2 \|x\|_{\mathcal{X}}^2$$
$$\leq \mathbb{E} C^2 \frac{1}{r^2} \|y - x\|^2 + C^2 \|x\|_{\mathcal{X}}^2$$
$$= \frac{C^2}{r^2}\sigma^2 d + C^2 \|x\|_{\mathcal{X}}^2 .$$

Furthermore

$$\mathbb{E}_y f_0(y)^2 \leq \mathbb{E}_y C^4 \|y\|_{\mathcal{X}}^4 \leq 8C^4 \mathbb{E} \left( \|x\|_{\mathcal{X}}^4 + \|y - x\|_{\mathcal{X}}^4 \right)$$
$$\leq 8C^4 \left( \|x\|_{\mathcal{X}}^4 + \frac{1}{r^4} \mathbb{E} \|y - x\|^4 \right)$$
$$\leq 8C^4 \left( \|x\|_{\mathcal{X}}^4 + \frac{4}{r^4} d\sigma^4 \right).$$

$\square$

**Lemma 11** (Norm comparison). *The $\|.\|_{\mathcal{X}}$ can be upper bounded by the Euclidean norm $\|.\|$ as*

$$\forall y \in \mathbb{R}^d, \frac{1}{R} \|y\| \leq \|y\|_{\mathcal{X}} \leq \frac{1}{r} \|y\|.$$

*Proof.* Note that for any $y \in \mathbb{R}^d$, for $\alpha = \|y\|/r$ we have $y/\alpha \in \mathcal{X}$. Therefore, from the definition of $\|.\|_{\mathcal{X}}$:

$$\|y\|_{\mathcal{X}} = \inf\{\alpha > 0, \frac{y}{\alpha} \in \mathcal{X}\} \leq \frac{\|y\|}{r}.$$

Furthermore, for $\alpha < \frac{\|y\|}{R}$, then $\left\|\frac{y}{\alpha}\right\| > R$, which means $y \notin \mathcal{X}$ (since $\mathcal{X}$ is contained in a ball of radius $R$). Therefore, $\|y\|_{\mathcal{X}} \geq \frac{\|y\|}{R}$. $\qquad\square$

**Lemma 12** (Gaussian smoothing). *For arbitrary unit direction $v$, given the smooth regularizer defined in equation 17 we have*

$$|Df(x)[v]| \leq \frac{1}{\sigma} \sqrt{\mathbb{E} f_0(y)^2}$$

$$|D^2 f[v,v]| \leq \frac{4}{\sigma^2} \sqrt{\mathbb{E} f_0(y)^2},$$

$$D^3 f(x)[v,w,u] \leq \frac{5}{\sigma^3} \sqrt{\mathbb{E} f_0(y)^2}.$$

*Proof.* Consider the function $f_0(y) e^{-\frac{(y-x)^2}{2\sigma^2}}$; it is continuous in both $y, x$ due to continuity of $f_0$ by Lemma 7, and its partial derivative with respect to $x$ in direction $v$ is $f_0(y)\langle \frac{y-x}{\sigma^2}, v\rangle$ which is again continuous wrt $x$ and $y$. Therefore, from the Leibnitz rule, for arbitrary direction $v$, $Df(x)[v]$ exists and is equal to

$$Df(x)[v] = \mathbb{E}_y \langle \frac{y-x}{\sigma^2}, v\rangle f_0(y).$$

Therefore, from Cauchy Schwarz

$$|Df(x)[v]| \leq \frac{1}{\sigma^2} \sqrt{\mathbb{E}\langle y-x, v\rangle^2} \sqrt{\mathbb{E} f_0(y)^2} = \frac{1}{\sigma} \sqrt{\mathbb{E} f_0(y)^2}.$$

For the second derivative

$$D^2 f(x)[v,w] = \mathbb{E}_y \left( \langle \frac{y-x}{\sigma^2}, v\rangle \langle \frac{y-x}{\sigma^2}, w\rangle f_0(y) - \frac{1}{\sigma^2} \langle v, w\rangle f_0(y) \right)$$

which gives

$$|D^2 f(x)[v,w]| \leq \left( \frac{1}{\sigma^2} \sqrt{\mathbb{E}_{\eta \sim N(0,1)} \eta^4} + \frac{1}{\sigma^2} \right) \sqrt{\mathbb{E} f_0(y)^2} = \frac{4}{\sigma^2} \sqrt{\mathbb{E} f_0(y)^2}.$$

where $\eta$ is normal gaussian with variance one. Similarly for the third derivative

$$D^3 f(x)[v,w,u] = \mathbb{E}_y \left( \langle \frac{y-x}{\sigma^2}, v\rangle \langle \frac{y-x}{\sigma^2}, w\rangle \langle \frac{y-x}{\sigma^2}, u\rangle f_0(y) - \frac{1}{\sigma^2} \sum_{u,v,w} \langle v, w\rangle \langle \frac{y-x}{\sigma^2}, u\rangle f_0(y) \right).$$

Therefore,

$$|D^3 f(x)[v,w,u]| \leq \left( \frac{1}{\sigma^3} \left(\mathbb{E}\eta^6\right)^{1/2} + \frac{1}{\sigma^3} \sqrt{\mathbb{E}\eta^2} \right) \sqrt{\mathbb{E} f_0(y)^2}$$

$$= \frac{1}{\sigma^3}(\sqrt{15} + 1)\sqrt{\mathbb{E} f_0(y)^2} \leq \frac{5}{\sigma^3} \sqrt{\mathbb{E} f_0(y)^2}.$$

$\qquad\square$

**Corollary 1** (Final smoothed derivatives). *For the smoothed barrier defined in Equation equation 17 and $x \in \mathcal{X}$, we have*

$$|f(x)| \le C^2 (\frac{\sigma^2}{r^2} d + 1)$$

$$|Df(x)[v]| \le \frac{C^2}{\sigma} \sqrt{8 \left( 1 + 4\frac{1}{r^4} d\sigma^4 \right)}$$

$$|D^2 f[v, v]| \le \frac{4C^2}{\sigma^2} \sqrt{8 \left( 1 + 4\frac{1}{r^4} d\sigma^4 \right)},$$

$$D^3 f(x)[v, w, u] \le \frac{5C^2}{\sigma^3} \sqrt{8 \left( 1 + 4\frac{1}{r^4} d\sigma^4 \right)}.$$

*Proof.* Directly by combining Lemmas 10 and 12. $\square$

**Theorem 8** (Existence of a smooth regularizer). *Given that there exists a 2-homogeneous regularizer $f_0 : \mathbb{R}^d \to \mathbb{R}$ that is $\alpha$-strongly convex w.r.t $\|.\|_{\mathcal{L}^c}$ and that $\max_{x \in \mathcal{X}} |f_0(x)| \le C^2$, then there also exists a smooth regularizer $f$ which is $\alpha$-strongly convex w.r.t $\|.\|_{\mathcal{L}^c}$ and*

$$|f(x)| = O(C^2),$$

$$|Df(x)[v]| = O(C^2 \frac{d^{1/4}}{r}),$$

$$|D^2 f[v, v]| = O(C^2 \frac{d^{1/2}}{r^2}),$$

$$|D^3 f(x)[v, w, u]| = O(C^2 \frac{d^{3/4}}{r^3}).$$

*Proof.* It is enough to set $\sigma = \frac{r}{d^{1/4}}$ in Corollary equation 1. $\square$

## C  CALCULATING THE REGULARIZER

In this section, building upon the properties that we showed for feasible points of the program 9, we show how to compute a suitable regularizer $g^{(\mathcal{I}^o)}$ on $\mathcal{X}$. To do so, we build a separation oracle for $P_{\mathcal{I}}$. We start by defining the notions of separation oracle, as well as membership and linear optimization oracle. Before defining these oracle, we need to state the definition of set neighborhoods.

**Definition 5** (Membership Oracle). *For convex set $\mathcal{D} \in \mathbb{R}^d$, a membership oracle receives a vector $y \in \mathbb{R}^d$ and real number $\delta > 0$ and with probability $1 - \delta$ asserts $y \in B(\mathcal{D}, \delta)$, or it asserts $y \notin B(\mathcal{D}, -\delta)$. We denote the computational cost of a query to our membership oracle by $\text{MEM}_{\mathcal{X}}(\delta)$.*

**Definition 6** (Set neighborhoods). *For a subset $\mathcal{D} \subseteq \mathbb{R}^d$, let $B(\mathcal{D}, \delta)$ be the set of points that are within distance $\delta$ of $\mathcal{D}$, and $B(\mathcal{D}, -\delta)$ be the set of points that where a ball of radius $\delta$ around them is completely included in $\mathcal{D}$.*

**Definition 7** (Separation Oracle). *For a convex set $\mathcal{L} \subseteq \mathbb{R}^d$, a separation oracle receives a vector $y \in \mathbb{R}^d$ and real number $\delta > 0$ and either asserts $y \in B(\mathcal{L}, \delta)$, or it returns a unit vector $c \in \mathbb{R}^d$ such that $c^\top y \le c^\top x + \delta$ for all $x \in B(\mathcal{L}, -\delta)$. We denote the computation time of separation oracle by $\text{SEP}_{\mathcal{L}}(\delta)$.*

**Definition 8** (Linear Optimization Oracle). *For a convex set $\mathcal{L} \subset \mathbb{R}^d$, a linear optimization oracle receives a unit vector $c \in \mathbb{R}^d$ and real number $\delta_{lin}$ and returns a point $y \in \mathcal{C}$ such that $\forall x \in \mathcal{C}$, $c^\top y \le c^\top x + \delta_{lin}$. We denote the computational cost of calling the linear optimization oracle by $\text{LINO}_{\mathcal{L}}(\delta_{lin})$.*

Separation, Membership, and Linear Optimization oracles are known to be equivalent and can be used to implement convex optimization over convex sets. Grötschel et al. (2012) Next, we state a simplified version of Theorem 42 in Lee et al. (2018) (or Theorem 15 in Lee et al. (2015)) on how to build a linear optimization oracle from a separation oracle for a convex set, which we use in the proof of Theorem 10.

**Theorem 9** (Theorem 15 in Lee et al. (2018) or Theorem 42 in Lee et al. (2015)). *Let $K$ be a convex set satisfying $B_2(0, r) \subset K \subset B_2(0, 1)$ and let $\kappa = \frac{1}{r}$. For $0 \le \epsilon < 1$, with probability $1 - \epsilon$, we can compute $x \in B(K, \epsilon)$ such that*

$$c^\top x \le \min_{x \in K} c^\top x + \epsilon \|c\|_2$$

*with an expected running time of $O\left(nSEP_\delta(K)\log(\frac{n\kappa}{\epsilon}) + n^3 \log^{O(1)}\left(\frac{n\kappa}{\epsilon}\right)\right)$, where $\delta = \left(\frac{\epsilon}{n\kappa}\right)^{\Theta(1)}$.*

Next, we state how we solve the optimization problem in Theorem equation 9 based on a separation oracle that we build for its feasibility set $P_{\mathcal{I}}$ in Section E.

**Theorem 10** (Computing the Regularizer - abstract). *In the context of Lemma 4 Then, given arbitrary accuracy parameter $0 < \epsilon_1 < 1$, there is a cutting-plane method that approximately solves the program in equation 9 and obtains an almost feasible instance $\mathcal{I}^o$, in the sense that*

    *1. $\max_{x \in \mathcal{X}} |g^{(\mathcal{I}^o)}(x)| \le C^2 + \gamma_2 d\tilde{c}_1 \epsilon + \epsilon_1$*

    *2. $g^{(\mathcal{I}^o)}(x)$ is $\alpha/4$ strongly convex with respect to $\|.\|_{\mathcal{L}^c}$,*

*and runs in time (assuming $N \ge d$)*

$$O\left(\left(\frac{N\left(C_0{}^2 + c_1^2 + c_2^2\right)(c_2 \vee 1)R}{\epsilon_1 \epsilon Lr}\right)^{O(d)}\left(\text{LINO}_{\mathcal{L}}\left(\frac{(r \wedge 1)}{R^2 \alpha}\left(\frac{\epsilon_1 \bar{\epsilon} L}{N\left(C_0{}^2 + c_0^2 + c_2^2\right)}\right)^{\Theta(1)}\right)\right)\right).$$

*Proof.* The program equation 9 is a linear optimization problem over the convex set $P_{\mathcal{I}}$, for which we can exploit the separation oracle that we constructed in Lemma 13. In particular, the result directly follows from a simplified version of Theorem 42 in Lee et al. (2015) (or Theorem 15 in Lee et al. (2018)), a classical result on how to build a linear optimization oracle from the separation oracle for a convex set. For convenience of the reader, we have restated this result in Theorem 9. According to this theorem, for any $0 < \epsilon_1 < 1$, with probability $1 - \epsilon_1$ we can compute an instance $\mathcal{I}^o$ such that its corresponding barrier $g^{(\mathcal{I}^o)}$ satisfies

    1. $\max_{x \in \mathcal{X}} |g^{(\mathcal{I}^o)}(x)| \le \max_{x \in \mathcal{X}} |g^{(\mathcal{I}^*)}(x)| + \epsilon_1$, where $\mathcal{I}^*$ is the optimal solution to the LP.

    2. $\mathcal{I}^o$ is $\epsilon_1$ close to a feasible instance $\mathcal{I}^{(r)}$ in Euclidean distance.

Now applying Lemma 4 we conclude the first argument, namely $\max_{x \in \mathcal{X}} |g^{(\mathcal{I}^o)}(x)| \le C^2 + \gamma_2 d\tilde{c}_1 \epsilon + \epsilon_1$. Now we need to show that $g^{(\mathcal{I}^o)}$ roughly remains $\Omega(\alpha)$ strongly convex w.r.t $\|.\|_{\mathcal{L}^c}$. For this, note that given $x \in \mathcal{X}$, if $\|x_j - x\| \ge \gamma \left(\frac{\epsilon\sqrt{d}c_0}{L}\right)^{1/3}$ and $\|x_i - x\| \le \epsilon$, then from Lemma 2 and the feasibility of $\mathcal{I}^{(r)}$ we have $r_{x_i} > g_{x_j}^{(\mathcal{I}^r)}(x) + \sqrt{d}c_0\epsilon$ where $r_{x_i}$ is the variable of the valid instance $\mathcal{I}^r$. But picking $\epsilon_1 \le \frac{\sqrt{d}c_0\epsilon}{2R^2}$ we get that $g_{x_i}^{(\mathcal{I}^o)}(x) > g_{x_j}^{(\mathcal{I}^o)}(x)$. Therefore, again the maximum at $x$ is achieved by one of the functions $g_{x_j}^{(\mathcal{I}^o)}(x)$ where $x_j$ is not farther than $\gamma \left(\frac{\epsilon\sqrt{d}c_0}{L}\right)^{1/3}$ of $x$. But then similar to Equation equation 55 in Lemma 14, for all $\hat{i} \in I$ and arbitrary direction $v$:

$$v^\top \nabla^2 g_{x_{\hat{i}}}^{(\mathcal{I}^r)}(x)v \ge \frac{\alpha}{2}\|v\|_{\mathcal{L}^c}^2.$$

On the other hand, $\|\mathcal{I}^o - \mathcal{I}^r\| \le \epsilon_1$ implies $\left\|\nabla^2 g_{x_{\hat{i}}}^{(\mathcal{I}^r)}(x) - \nabla^2 g_{x_{\hat{i}}}^{(\mathcal{I}^o)}(x)\right\|_F \le \epsilon_1$. Therefore, using $\epsilon_1 \le \frac{\alpha}{4r^2}$ we conclude

$$v^\top \nabla^2 g_{x_{\hat{i}}}^{(\mathcal{I}^o)}(x)v \ge \frac{\alpha}{4}\|v\|_{\mathcal{L}^c}^2,$$

which is the desired property. Finally, using the third argument in Lemma 4, we have the following runtime based on Theorem 9:

$$O\left(N \cdot \text{SEP}_{P_{\mathcal{I}}}(\delta)\log\left(\frac{1}{\delta}\right) + N^3 \log^{O(1)}\left(\frac{1}{\delta}\right)\right),$$

for

$$\delta \triangleq \left( \frac{\epsilon_1 L \bar{\epsilon}^3}{N\sqrt{(N+1)C_0{}^2 + Nd\left(c_0^2 + c_2^2\right)}} \right)^{\Theta(1)} = \left( \frac{\epsilon_1 \bar{\epsilon} L}{N\left(C_0{}^2 + c_0^2 + c_2^2\right)} \right)^{\Theta(1)}.$$

Note that from Lemma 13, for this choice of $\delta$ we have

$$\mathrm{SEP}_{P_\mathcal{I}}(\delta) = O\left( \frac{N\left(C_0{}^2 + c_1^2 + c_2^2\right)c_2 R}{\epsilon_1 \epsilon L r} \right)^{O(d)} \left( \mathrm{LINO}_\mathcal{L}\left( \frac{(r \wedge 1)}{R^2 \alpha} \left( \frac{\epsilon_1 \bar{\epsilon} L}{N\left(C_0{}^2 + c_0^2 + c_2^2\right)} \right)^{\Theta(1)} \right) + d^2 \right) + O\left(N^2 d^2\right),$$

which completes the proof. $\qquad\square$

Next, we appropriately instantiate the constants of the convex program equation 9 based on Theorem 8 and Lemma 4 in Theorem 1 below. We find the running time of our cutting-plane method to solve this program based on Theorem 10.

**Theorem 11** (Restatement of Theorem 1). *Assuming $R > 1, r < 1$ for simplicity, given that the best achievable rate for online linear optimization with action and constraint sets $(\mathcal{X}, \mathcal{L})$ is $O(\mathrm{Rate}(\mathcal{X}, \mathcal{L})\sqrt{T})$, there exists an algorithm that runs in time*

$$\left( \frac{dR}{r} \right)^{O(d^2)} \left( \mathrm{LINO}_\mathcal{L}\left( \left( \frac{r}{dR} \right)^{\Theta(d)} \right) \right),$$

*and calculates a regularizer $g^{(\mathcal{I}^o)}$ given by the representation $(\boldsymbol{\Sigma}, \mathbf{v}, \mathbf{r})$ as described in Section equation 7, which satisfies*

1. *$\sup_{x \in \mathcal{X}} |g^{(\mathcal{I}^o)}| \le 2\mathrm{Rate}(\mathcal{X}, \mathcal{L})^2$*

2. *$g^{(\mathcal{I}^o)}$ is 1-strongly convex w.r.t $\|.\|_{\mathcal{L}^c}$.*

*Proof.* Let $C \triangleq \mathrm{Rate}(\mathcal{X}, \mathcal{L})$. From Theorem 8 there exists a 2-homogeneous barrier which is $\tilde{c}_1 = O(C^2 \frac{d^{1/4}}{r})$ Lipschitz, $\tilde{c}_2 = O(C^2 \frac{d^{1/2}}{r^2})$ Gradient Lipschitz, $L = O(C^2 \frac{d^{3/4}}{r^3})$ Hessian Lipschitz, and 1-strongly convex w.r.t $\|.\|_{\mathcal{L}^c}$. Therefore, to enjoy the properties of Lemma 4, assuming that we guarantee,

$$\bar{\epsilon}^3 \le \min\{\frac{\tilde{c}_1}{L}, \frac{\tilde{c}_2}{L}, \frac{C^2}{L}\} \tag{19}$$

then we get that $c_0, c_2, C_0$ are of the same order as $\tilde{c}_1, \tilde{c}_2, C^2$, respectively (this follows from the definition of $c_0, c_2, C_0$ which involves the term $L\bar{\epsilon}^3$). Now following the condition of Lemma 4, we consider a cover of accuracy $\epsilon$ such that

$$\epsilon \le \min\{\frac{1}{r^2}, \frac{r^6}{C^6 d^2}, \frac{r}{d^{1/4}}, C\frac{d^{3/8}}{r^{1/2}}, rd^{1/4}, \frac{r^7}{R^6 C^6 d^{11/8}}\}.$$

where we set $L = \gamma_5 C^2 \frac{d^{3/4}}{r^3}$ for small enough constant $\gamma_5$. For simplicity if either $R$ or $C$ were smaller than one, we upper bound them by one, so we can assume $R, C \ge 1$ without loss of generality. Similarly if $r < 1$, we can take $r = 1$, so without loss of generality we assume $r = 1$. Then, the above bound simplifies to

$$\epsilon \le \frac{r^6}{R^6 C^6 d^2}. \tag{20}$$

Furthermore we consider the discretization set $\tilde{\mathcal{X}}$ to be points each entry is of the form $k\bar{\epsilon}$ for an integer $k$. Then, to guarantee equation 19 we should have

$$\bar{\epsilon}^3 \le \frac{r^2}{d^{1/2}}. \tag{21}$$

On the other hand, rounding every point $x$ to its closest multiple of $\bar{\epsilon}$ in each coordinate implies that the cover has accuracy as small as $\epsilon = \sqrt{d}\bar{\epsilon}$. Hence, to satisfy condition equation 20 we set

$$\bar{\epsilon} \triangleq \frac{\gamma_4 r^6}{R^6 C^6 d^2 \sqrt{d}},$$

$$\epsilon \triangleq \frac{\gamma_4 r^6}{R^6 C^6 d^2},$$

for small enough constant $\gamma_4$. Then, it is easy to check that condition equation 21 is automatically satisfied. Furthermore, with this choice of $\bar{\epsilon}$ we see that $\gamma_2 d \tilde{c}_1 \epsilon \leq \frac{C^2}{2}$ for small enough constant $\gamma_4$ ($\gamma_2$ is defined in Lemma 10); hence, from the guarantee of Lemma 10

$$\max_{x \in \mathcal{X}} |g^{(\mathcal{I}^o)}(x)| \leq C^2 + \gamma_2 d \tilde{c}_1 \epsilon + \epsilon_1 \leq \frac{3}{2} C^2 + \epsilon_1,$$

where recall $\epsilon_1$ is the accuracy parameter for our solver in Lemma 10. Setting

$$\epsilon_1 = \frac{C^2}{2},$$

we conclude

$$\max_{x \in \mathcal{X}} |g^{(\mathcal{I}^o)}(x)| \leq 2C^2.$$

Note that the attained constant two behind $C^2$ does not matter since the parameter $C$ of the smoothed barrier in Theorem 8 can be off by a universal constant from $\mathrm{Rate}(\mathcal{X}, \mathcal{L})$. Now since the regularizer $\tilde{f}$ is $\alpha = 1$ strongly convex, Lemma 10 also guarantees that the regularizer that we find, $g^{(\mathcal{I}^o)}(x)$, is $\frac{1}{4}$ strongly-convex with respect to $\|.\|_{\mathcal{L}^c}$. Finally from the runtime guarantee of Lemma 10, finding such regularizer has runtime

$$O \left( \frac{NR}{r} \right)^{O(d)} \left( \mathrm{LINO}_{\mathcal{L}} \left( \left( \frac{r}{NR} \right)^{\Theta(1)} \right) \right),$$

where we used the fact that $C_0^2 + c_1^2 + c_2^2 = O(C^4 R^4 d^2)$ and $d \leq N$, and that we can upper bound $C$ by $R$ (Note that we dropped the $d$ in the term $\frac{NRd}{r}$ since $N$ is already exponentially large in $d$). Furthermore, the cover that we considered has size at most $N = |\tilde{\mathcal{X}}| = O\left(\frac{R}{\epsilon}\right)^d = \left(\frac{dR}{r}\right)^{O(d)}$. Therefore, the overall runtime is

$$\left( \frac{dR}{r} \right)^{O(d^2)} \left( \mathrm{LINO}_{\mathcal{L}} \left( \left( \frac{r}{dR} \right)^{\Theta(d)} \right) \right).$$

$\square$

# D  ONLINE LINEAR OPTIMIZATION

Here we show how to run FTRL with regularizer $g^{\mathcal{I}^o}$ that is based on the instance $\mathcal{I}^o$ which we computed in Section C for a general instance of the online linear optimization problem as we defined in Section 3.1; as we mentioned, our approach results in the optimal information theoretic rate up to universal constants.

**Theorem 12** (Optimal online optimization). *Consider the problem of online linear optimization with action and loss sets $(\mathcal{X}, \mathcal{L})$ as described in Section 3.1. Given access to the regularizer $g^{\mathcal{I}^o}$ for the instance $\mathcal{I}^o$ of the program 9 that we can compute as described in Theorem 1 and a membership oracle for $\mathcal{X}$, there is a cutting-plane algorithm to run FTRL with regularizer $g^{\mathcal{I}^o}$, with running time*

$$O \left( T d^2 \ln^{O(1)} (dRT) (\mathrm{MEM}_{\mathcal{X}}(\delta) + 1) \right),$$

*which guarantees regret $O(\mathrm{Rate}(\mathcal{X}, \mathcal{L})\sqrt{T})$.*

*Proof.* We run FTRL with the regularizer $g^{(\mathcal{I}^o)}$; namely, to calculate each step $1 \le t \le T$, we solve the following convex optimization using separation oracle for $\mathcal{X}$:

$$x_t = \arg\min_{x \in \mathcal{X}} G_t(x) \tag{22}$$

$$G_t(x) \triangleq \langle x, \sum_{s=1}^{t-1} g_s \rangle + g^{\mathcal{I}^o}(x), \tag{23}$$

up to accuracy $O(\frac{\alpha r}{R^2 T})$, namely for $\tilde{x}_t$ being the output of the algorithm we have

$$G_t(\tilde{x}_t) - G_t(x_t) \le O(\frac{\alpha r}{R^2 T}) | \sup_{x \in \mathcal{X}} G_t(x) - \inf_{x \in \mathcal{X}} G_t(x) | = O(\frac{\alpha r}{R^2 T} \mathrm{Rate}(\mathcal{X}, \mathcal{L})^2). \tag{24}$$

Note that we used the property that for the regularizer $g^{\mathcal{I}^o}$ that we calculate in Theorem 10 we have $\sup_{x \in \mathcal{X}} |g^{(\mathcal{I}^o)}| \le 2\mathrm{Rate}(\mathcal{X}, \mathcal{L})^2$. Then, from Theorem 1 in Lee et al. (2018), there is a cutting-plane method whose number of queries to a membership oracle for $\mathcal{X}$ is

$$O\left( d^2 \ln^{O(1)} (dRT) \right)$$

in addition to $O\left( d^2 \ln^{O(1)} (dRT) \right)$ arithmetic operations.

But since $x_t$ is the global minimizer of $G_t$ we have $\nabla G_t(x_t) = 0$, and further from $\alpha/4$ strong convexity of $G_t$ w.r.t. $\|.\|_{\mathcal{L}^c}$:

$$G_t(\tilde{x}_t) - G_t(x_t) \ge \frac{\alpha}{4} \|x_t - \tilde{x}_t\|_{\mathcal{L}^c}^2 \ge \frac{\alpha r}{4} \|x_t - \tilde{x}_t\|^2,$$

which combined with equation 24 implies

$$\|x_t - \tilde{x}_t\| \le \frac{\mathrm{Rate}(\mathcal{X}, \mathcal{L})}{R\sqrt{T}}.$$

Then, from the mirror descent guarantee we have the following regret bound for the sequence $x_t$

$$\mathbb{E}\left( \max_{x^* \in \mathcal{X}} \sum_{t=1}^{T} \langle x_t, g_t \rangle - \langle x^*, g_t \rangle \right) = O(\mathrm{Rate}(\mathcal{X}, \mathcal{L})\sqrt{T}). \tag{25}$$

On the other hand, using the fact that $\|g_t\| \le R$ and that $\mathcal{L} \subseteq B_R(0)$,

$$\mathbb{E}\left( \sum_{t=1}^{T} \langle x_t, g_t \rangle - \langle \tilde{x}_t, g_t \rangle \right)$$

$$\mathbb{E}\left( \sum_{t=1}^{T} \|x_t - \tilde{x}_t\| \|g_t\| \right)$$

$$\le \mathrm{Rate}(\mathcal{X}, \mathcal{L})\sqrt{T}. \tag{26}$$

Combining equation 25 and equation 26 completes the proof for the regret guarantee. $\qquad\square$

## E SEPARATION ORACLE

Here we show a separation oracle for the feasible polytope $P_\mathcal{I}$ of program 9.

**Lemma 13** (Linear optimization oracle for $\mathcal{L} \to$ Separation Oracle). *The polytope $P_\mathcal{I}$ for $\mathcal{I} = (\mathbf{r}, \mathbf{v}, \boldsymbol{\Sigma})$ defined in equation 9 has a separation oracle with computational cost*

$$\mathrm{SEP}_K(\delta) = O\left( \frac{2c_2 R^3}{\delta r^3} \right)^d \left( \mathrm{LINO}_\mathcal{L}\left( \delta (1 \wedge r)/(8\alpha R^2) \right) + d^2 \right) + O\left( |\tilde{\mathcal{X}}|^2 d^2 \right),$$

*where $\mathrm{LINO}_\mathcal{L}\left( \delta (1 \wedge r)/(8\alpha R^2) \right)$ is the cost of a linear optimization oracle for $\mathcal{L}$ with parameter $\delta = (1 \wedge r)/(8\alpha R^2)$.*

*Proof.* We can readily check if conditions (1) and (2) hold for the instance $\mathcal{I}$, and if not, that condition defines the direction $c$ for which $\langle \mathcal{I}, c \rangle \geq \langle \tilde{\mathcal{I}}, c \rangle$ for all $\tilde{\mathcal{I}} \in P_{\mathcal{I}}$. To check condition (3) we can do singular value decomposition in $O(d^3)$ Condition (4) is a bit trickier since it might be hard to directly maximize $v^\top \Sigma_{x_i} v$ over $\mathcal{L}$. Therefore, we work with the discretization set $\tilde{S}_d$ of the unit $d$-dimensional sphere; in particular, for every unit direction $\tilde{v} \in \tilde{S}_d$, we consider condition (5) with a margin $\delta_m$, namely

$$v^\top \Sigma_{x_i} v / \|v\|_{\mathcal{L}^c}^2 \geq \alpha(1 + \delta_m). \tag{27}$$

This margin allows us to easily obtain a feasible solution in $P_{\mathcal{I}}$ which satisfies $v^\top \Sigma_{x_i} v \geq \alpha$ for all $v \in \mathcal{L}$, using condition in equation 27 which is only for the discretization points; moreover, we check equation 27 with our linear optimization oracle which has error $\delta_{lin}$ in calculating $\|v\|_{\mathcal{L}^c}$; namely, suppose equation 27 holds for all $\tilde{v} \in \tilde{S}_d$ given that we substitute $\|v\|_{\mathcal{L}^c}$ in equation 27 with the output of $\text{LINO}_{\mathcal{L}}(\delta_{lin})$. Then, we are guaranteed that for every $\tilde{v} \in \tilde{S}^d$:

$$\tilde{v}^\top \Sigma_{x_i} \tilde{v} / \left(\text{LINO}_{\mathcal{L}}(\delta_{lin})[\tilde{v}]\right)^2 \geq \alpha(1 + \delta_m). \tag{28}$$

Now from the fact that $\|\tilde{v}\|_{\mathcal{L}^c} \geq r$ and $\text{LINO}_{\mathcal{L}}(\delta_{lin})[\tilde{v}] \geq \|\tilde{v}\|_{\mathcal{L}^c} - \delta_{lin}$, picking $\delta_{lin} \leq \frac{r\delta_m}{2}$, we get that

$$\tilde{v}^\top \Sigma_{x_i} \tilde{v} / \left((1 - \delta_{lin}/2)\|\tilde{v}\|_{\mathcal{L}^c}\right)^2 \geq \alpha(1 + \delta_m), \tag{29}$$

which using the fact that we picked $\delta_{lin} \leq \delta_m/4$ implies

$$\tilde{v}^\top \Sigma_{x_i} \tilde{v} / \left(\|\tilde{v}\|_{\mathcal{L}^c}\right)^2 \geq (1 - \delta_{lin}/2)^2 \alpha(1 + \delta_m) \geq \alpha(1 + \delta_m/2). \tag{30}$$

Now for arbitrary direction $v \in S^d$ on the unit sphere, we bound the value of the quadratic form the closest point in the discretization set: namely for $\tilde{v} \in \tilde{S}^d$ where $\|\tilde{v} - v\| \leq \tilde{\epsilon}$:

$$|v^\top \Sigma_{x_i} v / \|v\|_{\mathcal{L}^c}^2 - \tilde{v}^\top \Sigma_{x_i} \tilde{v} / \|\tilde{v}\|_{\mathcal{L}^c}^2|$$
$$= |v^\top \Sigma_{x_i} v / \|v\|_{\mathcal{L}^c}^2 - \tilde{v}^\top \Sigma_{x_i} \tilde{v} / \|v\|_{\mathcal{L}^c}^2| + |\tilde{v}^\top \Sigma_{x_i} \tilde{v} / \|v\|_{\mathcal{L}^c}^2 - \tilde{v}^\top \Sigma_{x_i} \tilde{v} / \|\tilde{v}\|_{\mathcal{L}^c}^2|. \tag{31}$$

but for the first term, using $\|\Sigma_{x_i}\| \leq c_2$:

$$|v^\top \Sigma_{x_i} v - \tilde{v}^\top \Sigma_{x_i} \tilde{v}| \leq |(v - \tilde{v})^\top \Sigma_{x_i} v| + |(v - \tilde{v})^\top \Sigma_{x_i} \tilde{v}| \qquad \leq 2c_2 \|v - \tilde{v}\| \leq 2c_2 \tilde{\epsilon}$$

and $\|v\|_{\mathcal{L}^c} \geq r$. Hence, from $\tilde{\epsilon} < 1$

$$|v^\top \Sigma_{x_i} v / \|v\|_{\mathcal{L}^c}^2 - \tilde{v}^\top \Sigma_{x_i} \tilde{v} / \|v\|_{\mathcal{L}^c}^2| \leq 2c_2 \frac{\tilde{\epsilon}}{r^2}. \tag{32}$$

For the second term, using the fact that $r \leq \|\tilde{v}\|_{\mathcal{L}^c}$, $\|v\|_{\mathcal{L}^c} \leq R$ and $\|\tilde{v} - v\|_{\mathcal{L}^c} \leq R \|\tilde{v} - v\|$:

$$|\tilde{v}^\top \Sigma_{x_i} \tilde{v} / \|v\|_{\mathcal{L}^c}^2 - \tilde{v}^\top \Sigma_{x_i} \tilde{v} / \|\tilde{v}\|_{\mathcal{L}^c}^2| \leq c_2 \frac{\|\tilde{v}\|_{\mathcal{L}^c}^2 - \|v\|_{\mathcal{L}^c}^2}{\|\tilde{v}\|_{\mathcal{L}^c}^2 \|v\|_{\mathcal{L}^c}^2}$$
$$\leq c_2 \frac{\|\tilde{v} - v\|_{\mathcal{L}^c}(\|v\|_{\mathcal{L}^c} + \|\tilde{v}\|_{\mathcal{L}^c})}{\|\tilde{v}\|_{\mathcal{L}^c}^2 \|v\|_{\mathcal{L}^c}^2}$$
$$= c_2 \frac{\|\tilde{v} - v\|_{\mathcal{L}^c}}{\|\tilde{v}\|_{\mathcal{L}^c} \|v\|_{\mathcal{L}^c}^2} + c_2 \frac{\|\tilde{v} - v\|_{\mathcal{L}^c}}{\|\tilde{v}\|_{\mathcal{L}^c}^2 \|v\|_{\mathcal{L}^c}}$$
$$\leq \frac{2\tilde{\epsilon} c_2 R}{r^3}. \tag{33}$$

Combining Equations equation 32 and equation 33 (from $R/r \geq 1$) and plugging into equation 31

$$|v^\top \Sigma_{x_i} v / \|v\|_{\mathcal{L}^c}^2 - \tilde{v}^\top \Sigma_{x_i} \tilde{v} / \|\tilde{v}\|_{\mathcal{L}^c}^2| \leq \frac{4\tilde{\epsilon} c_2 R}{r^3},$$

which combined with equation 30 and triangle inequality

$$v^\top \Sigma_{x_i} v / \left(\|v\|_{\mathcal{L}^c}\right)^2 \geq \alpha(1 + \delta_m/2) - \frac{4\tilde{\epsilon} c_2 R}{r^3}.$$

Using $\tilde{\epsilon} \le \frac{\alpha r^3 \delta_m}{c_2 R}$, we get

$$v^\top \Sigma_{x_i} v / \left( \|v\|_{\mathcal{L}^c} \right)^2 \ge \alpha(1 + \delta_m/4).$$

Recall that $v$ was arbitrary in $S^d$. Therefore, in the case when all inequalities in equation 28 are satisfied, we showed that $\mathcal{I}$ indeed satisfies condition (4) in equation 9. Finally if any of the inequalities equation 29 are violated, i.e. if $\tilde{v}^\top \Sigma_{x_i} \tilde{v} / \left( \text{LINO}_{\mathcal{L}} \left( \delta_{lin} \right) [\tilde{v}] \right)^2 \ge \alpha(1 + \delta_m)$, then similar to equation 29 we get

$$\tilde{v}^\top \Sigma_{x_i} \tilde{v} / \left( (1 + \delta_{lin}/2) \|\tilde{v}\|_{\mathcal{L}^c} \right)^2 \le \alpha(1 + \delta_m) \le \tilde{v}^\top \Sigma_{x_i} \tilde{v} / \left( \text{LINO}_{\mathcal{L}} \left( \delta_{lin} \right) [\tilde{v}] \right)^2 \le \alpha(1 + \delta_m),$$

which implies (from $\delta_{lin} \le \delta_m/4$)

$$\tilde{v}^\top \Sigma_{x_i} \tilde{v} / \left( \|\tilde{v}\|_{\mathcal{L}^c} \right)^2 \le \alpha \left( 1 + \delta_{lin}/2 \right)^2 (1 + \delta_m) \le \alpha(1 + 2\delta_m).$$

Therefore, we find that the unit direction $\tilde{v}\tilde{v}^\top$ which satisfies

$$\langle \tilde{v}\tilde{v}^\top, \Sigma_{x_i} \rangle \le \alpha \|\tilde{v}\|_{\mathcal{L}^c}^2 + 2\alpha\delta_m \|\tilde{v}\|_{\mathcal{L}^c}^2$$
$$\le \alpha \|\tilde{v}\|_{\mathcal{L}^c}^2 + 2\alpha\delta_m R^2,$$

while for a valid $\mathcal{I} \in \mathcal{P}_{\mathcal{I}}$, we should have $\langle vv^\top, \Sigma_{x_i} \rangle \ge \alpha \|v\|_{\mathcal{L}^c}^2$ for all unit directions $v$. Hence, we constructed a separation oracle with $2\alpha\delta_m R^2$, which uses $|\tilde{S}^d|$ queries to the linear optimization oracle, and its overal computational cost is $O\left( |\tilde{S}^d| \left( \text{LINO}_{\mathcal{L}} \left( \delta_{lin} \right) + d^2 \right) + |\tilde{\mathcal{X}}|^2 d^2 \right)$. Finally to have a $\delta$-separation oracle, we need to guarantee $2\alpha\delta_m R^2 \le \delta$, $\delta_{lin} \le \frac{\delta_m}{4} \wedge \frac{r\delta_m}{2}$, $\tilde{\epsilon} \le \frac{\alpha r^3 \delta_m}{c_2 R}$, hence we pick

$$\delta_m \triangleq \frac{\delta}{2\alpha R^2},$$
$$\delta_{lin} \triangleq \frac{\delta_m (1 \wedge r)}{4} = \frac{\delta (1 \wedge r)}{8\alpha R^2},$$
$$\tilde{\epsilon} \triangleq \frac{r^3 \delta}{2c_2 R^3}.$$

Hence, the overall computational cost is

$$O \left( 1/\tilde{\epsilon} \right)^d \left( \text{LINO}_{\mathcal{L}} \left( \delta \left( 1 \wedge r \right) / (8\alpha R^2) \right) + d^2 \right) + O\left( |\tilde{\mathcal{X}}|^2 d^2 \right)$$
$$= O \left( \frac{2c_2 R^3}{\delta r^3} \right)^d \left( \text{LINO}_{\mathcal{L}} \left( \delta \left( 1 \wedge r \right) / (8\alpha R^2) \right) + d^2 \right) + O\left( |\tilde{\mathcal{X}}|^2 d^2 \right).$$

$\square$

# F PROOFS FOR SECTIONS 5 AND 7

## F.1 PROOF OF LEMMA 1

For the lower bound, we use the inequality $\nabla^2 f(x_1) \succcurlyeq \nabla^2 f(x_0) - L \|x_1 - x_0\| I$:

$$f(x) = f(x_0) + \langle \nabla f(x_0), x - x_0 \rangle + \int_0^1 \int_0^t (x - x_0)^\top \nabla^2 f(x_0 + s(x - x_0))(x - x_0) ds dt$$
$$\ge f(x_0) + \langle \nabla f(x_0), x - x_0 \rangle + \int_0^1 \int_0^t (x - x_0)^\top \left( \nabla^2 f(x_0) - sL\|x - x_0\|I \right) (x - x_0) ds dt$$
$$= f(x_0) + \langle \nabla f(x_0), x - x_0 \rangle + \frac{1}{2}(x - x_0)^\top \nabla^2 f(x_0)(x - x_0) - \frac{L}{6} \|x - x_0\|^3$$
$$= f_{x_0}(x) + \frac{L}{6} \|x - x_0\|^3.$$

For upper bound, we use the inequality $\nabla^2 f(x_1) \preccurlyeq \nabla^2 f(x_0) + L \|x_1 - x_0\| I$:

$$f(x) \le f(x_0) + \langle \nabla f(x_0), x - x_0 \rangle + \int_0^1 \int_0^t (x - x_0)^\top (\nabla^2 f(x_0) + sL\|x - x_0\|I)(x - x_0) ds dt$$

$$= f(x_0) + \langle \nabla f(x_0), x - x_0 \rangle + \frac{1}{2}(x - x_0)^\top \nabla^2 f(x_0)(x - x_0) + \frac{L}{6}\|x - x_0\|^3$$

$$= f_{x_0}(x) + \frac{L}{2}\|x - x_0\|^3.$$

### F.2 PROOF OF LEMMA 2

We denote $g_{x_i}^{(\mathcal{I})}(x)$ in short by $g_{x_i}(x)$, and without loss of generality let $x_i = x_0$ and $x_j = x_1$. First, note that we can translate the convex program conditions on the norm of $v_{x_i}$ to

$$\|v_{x_i}\| \le c_1,$$

for $c_1 = \sqrt{d}c_0$. From the program constraint we have

$$g_{x_1}(x_0) + \frac{15L}{96}\|x_1 - x_0\|^3 \le r_{x_0}. \tag{34}$$

On the other hand, from $\|x_0 - x\| \le \epsilon$ and the norm bounds on gradient and Hessian

$$|g_{x_1}(x_0) - g_{x_1}(x)| \le |v_{x_1}^\top (x_0 - x)| + |(x_0 - x)^\top \Sigma_{x_0} (x_0 + x - 2x_1)| + \frac{L}{3}|\|x_1 - x_0\|^3 - \|x_1 - x\|^3| \tag{35}$$

$$\le c_1 \|x_0 - x\| + c_2 \|x_0 - x\| (2\|x_0 - x_1\| + \|x_0 - x\|) \tag{36}$$

$$+ \frac{L}{3}\|x_0 - x\| \left( \|x_1 - x_0\|^2 + \|x_1 - x\|^2 + \|x_1 - x_0\| \|x_1 - x\| \right), \tag{37}$$

$$\le c_1 \|x_0 - x\| + c_2 \|x_0 - x\| (2\|x_0 - x_1\| + \|x_0 - x\|) \tag{38}$$

$$+ \frac{L}{3}\|x_0 - x\| \left( 4\|x_1 - x_0\|^2 + 2\|x_0 - x\|^2 \right), \tag{39}$$

where in the last line we used

$$\|x_1 - x_0\|^2 + \|x_1 - x\|^2 + \|x_1 - x_0\| \|x_1 - x\| \le 2\|x_1 - x_0\|^2 + 2\|x_1 - x\|^2 \tag{40}$$

$$\le 4\|x_1 - x_0\|^2 + 2\|x_0 - x\|^2. \tag{41}$$

Note that picking $\gamma \ge 3$, from the triangle inequality, $\|x - x_1\| \ge 3\left(\frac{\epsilon c_1}{L}\right)^{1/3}$, and the condition that $\frac{\epsilon\sqrt{d}c_0}{L} \le 1$,

$$\|x_0 - x_1\| \ge \|x_1 - x\| - \|x - x_0\| \ge 2\left(\frac{\epsilon c_1}{L}\right)^{1/3}. \tag{42}$$

Now based on equation 42, for the first term in equation 39, we can write

$$c_1 \|x_0 - x\| \le c_1 \epsilon \le \frac{L}{48}\|x_1 - x_0\|^3, \tag{43}$$

Similarly, also because $\epsilon \le \frac{L}{2000 c_1 c_2^3}$, for the second term we have

$$2c_2 \|x_0 - x\| \|x_0 - x_1\| \le \frac{L}{24}\|x_0 - x_1\|^3, \tag{44}$$

and because $\epsilon \le \frac{8L}{c_2}$,

$$2c_2 \|x_0 - x\|^2 \le \frac{L}{24}\|x_0 - x_1\|^3. \tag{45}$$

Finally for the last term, because $\epsilon \le \sqrt{\frac{c_1}{4096}}$,

$$\frac{4L}{3}\|x_0 - x\| \|x_1 - x_0\|^2 \le \frac{L}{48}\|x_0 - x_1\|^3 \tag{46}$$

and

$$\frac{4L}{3}\|x_0 - x\|^3 \leq \frac{L}{48}\|x_0 - x_1\|^3. \tag{47}$$

Therefore, defining

$$\psi_{x_0,x}(\|x_0 - x_1\|) \triangleq c_1\|x_0 - x\| + c_2\|x_0 - x\|(2\|x_0 - x_1\| + \|x_0 - x\|)$$
$$+ \frac{L}{3}\|x_0 - x\|\left(4\|x_1 - x_0\|^2 + 2\|x_0 - x\|^2\right),$$

we showed in equation 39 that for arbitrary $x_1$,

$$|g_{x_1}(x_0) - g_{x_1}(x)| \leq \psi_{x_0,x}(\|x_0 - x_1\|), \tag{48}$$

and for $x_1$ such that $\|x - x_1\| \geq 3\left(\frac{\epsilon\sqrt{d}c_0}{L}\right)^{1/3}$, or $\|x_1 - x_0\| \geq 2\left(\frac{\epsilon c_1}{L}\right)^{1/3}$, Combining equation 43, equation 44, equation 41, equation 46, equation 47 with equation 39:

$$\psi_{x_0,x}(\|x_0 - x_1\|) \leq \frac{3L}{48}\|x_0 - x_1\|^3. \tag{49}$$

Therefore, for $\|x - x_1\| \geq 4\left(\frac{\epsilon\sqrt{d}c_0}{L}\right)^{1/3}$,

$$|g_{x_1}(x_0) - g_{x_1}(x)| \leq \frac{7L}{48}\|x_0 - x_1\|^3,$$

which combined with Equation equation 34

$$g_{x_1}(x) + \frac{L}{96}\|x_1 - x_0\|^3 \leq r_{x_0}. \tag{50}$$

On the other hand, note that

$$|g_{x_0}(x) - r_{x_0}| \leq |v_{x_0}^\top(x - x_0)| + \frac{1}{2}(x - x_0)^\top\Sigma_{x_0}(x - x_0) \leq c_1\epsilon + \frac{c_2}{2}\epsilon^2 \leq 2c_1\epsilon,$$

where in the last line we used $\epsilon \leq \frac{c_1}{c_2}$. But now picking the constant $\gamma$ large enough we can guarantee that

$$\frac{L}{96}\|x_0 - x_1\|^3 \geq 3c_1\epsilon.$$

Combining equation 51 with equation 50, we conclude the first argument

$$g_{x_1}(x) + c_1\epsilon \leq g_{x_0}(x).$$

On the other hand, note that $\psi_{x_0,x}(x_1)$ is increasing in $\|x_1 - x_0\|$. Therefore, combining equation 48 and equation 49, for any $x_1$ such that $\|x_1 - x\| \leq \gamma\left(\frac{\epsilon\sqrt{d}c_0}{L}\right)^{1/3}$

$$|g_{x_1}(x_0) - g_{x_1}(x)| \leq \psi_{x_0,x}(\|x_0 - x_1\|) \leq \psi_{x_0,x}(\gamma\left(\frac{\epsilon\sqrt{d}c_0}{L}\right)^{1/3}) \leq \frac{3L}{48}\left(\left(\gamma\frac{\epsilon\sqrt{d}c_0}{L}\right)^{1/3}\right)^3 \tag{51}$$

$$= \gamma_2\epsilon\sqrt{d}c_0. \tag{52}$$

### F.3 PROOF OF LEMMA 3

Note that the Hessian of $\|x - x_0\|^3$ is $\alpha$ strong convexity of $f$ means for $v$ with $\|v\|_{\mathcal{L}} = 1$ we have $v^\top\nabla^2 f(x_0)v \geq \alpha$. But from Assumption equation 1 we get $\|v\| \leq R$. Therefore,

$$v^\top\nabla^2 f_{x_0}(x)v = v^\top\left(\nabla^2 f(x_0) - L\nabla(\|x - x_0\|(x - x_0))\right)v$$
$$= v^\top\left(\nabla^2 f(x_0) - L\nabla(\|x - x_0\|(x - x_0))\right)v$$
$$= v^\top\left(\nabla^2 f(x_0) - L\|x - x_0\|I - \frac{L}{\|x - x_0\|}(x - x_0)(x - x_0)^\top\right)v$$
$$\geq \alpha - 2R^2 L\|x - x_0\|$$
$$\geq \frac{\alpha}{2}.$$

### F.4 PROOF OF THEOREM 4

Here we prove Theorem 4. Before diving into the proof, we need to state and prove Lemma 14 so that we can obtain an $\alpha/2$ strong convexity property for the approximate regularizer in Theorem 4. In particular, Lemma 14 combines Lemmas 2 and 3 and concludes that the feasibility of $\mathcal{I}$ for the program implies strong convexity of $g$ with respect to $\|.\|_{\mathcal{L}^c}$.

**Lemma 14** (Program feasibility $\rightarrow$ strong convexity). *Suppose $\mathcal{I} = (\mathbf{r}, \mathbf{v}, \mathbf{\Sigma})$ is a feasible solution to LP equation 9 with respect to an $\epsilon$-cover $\{x_i\}_{i=1}^N$ in $\mathcal{X}$ for the Euclidean norm, i.e. $\forall x \in \mathcal{X}$, $\exists x_i$ s.t. $\|x - x_i\| \leq \epsilon$, where $\epsilon$ satisfies*

$$\epsilon \leq \frac{\alpha^3}{512R^6 L^2 c_0 \sqrt{d}}.$$

*Then, for any point $x \in \mathcal{X}$, $g$ is second order continuously right and left differentiable with*

$$D^{2,l}g(x)[v,v], D^{2,r}g(x)[v,v] \geq \frac{\alpha}{2} \|v\|_{\mathcal{L}^c}^2,$$

*where $D^{2,l}g(x)[v,v]$ and $D^{2,r}g(x)[v,v]$ denote the left and right second order directional derivative of $f$ at $x$ in direction $v$.*

*Proof.* For $x \in \mathcal{X}$ let $I(x) = \arg\max_{i \in [N]} g_{x_i}(x)$ be the set of indices for which $g_{x_i}(x)$ achieves its maximum at $x$. First, note that for the one-dimensional function $h(t) = g^{(\mathcal{I})}(x+tv)$, the subgradient of $h$ zero is exactly

$$[\min_{i \in I(x)} Dg_{x_i}(x)[v], \max_{i \in I(x)} Dg_{x_i}(x)[v]],$$

due to the convexity of $g_{x_i}$'s. In fact, $h'^l(0) = \min_{i \in I(x)} Dg_{x_i}(x)[v]$ and $h'^r(0) = \max_{i \in I(x)} Dg_{x_i}(x)[v]$. Now let

$$I^{r,v} = \arg\max_{i \in I(x)} Dg_{x_i}(x)[v]$$

$$I^{l,v} = \arg\min_{i \in I(x)} Dg_{x_i}(x)[v].$$

Then the second left and right directional derivatives at point $x$ are given by

$$D^{2,l}g(x)[v,v] = h''^l(0) = \max_{i \in I^l(x)} D^2 g_{x_i}(x)[v,v], \tag{53}$$

$$D^{2,l}g(x)[v,v] = h''^r(0) = \max_{i \in I^r(x)} D^2 g_{x_i}[v,v]. \tag{54}$$

Furthermore, note that from Lemma 2, for every $x_i$ such that $\|x_i - x\| \geq 4\left(\frac{\epsilon\sqrt{d}c_0}{L}\right)^{1/3}$, we have $g_{x_i}^{(\mathcal{I})}(x) < g_{x_0}^{(\mathcal{I})}(x)$, therefore $i \notin I$. Hence, we should have $\left\|x - x_{\hat{i}(x)}\right\| \leq 4\left(\frac{\epsilon\sqrt{d}c_0}{L}\right)^{1/3}$ for all $\hat{i} \in I$. But using the upper bound given on $\epsilon$ we get

$$\left\|x - x_{\hat{i}(x)}\right\| \leq 4\left(\frac{\epsilon\sqrt{d}c_0}{L}\right)^{1/3} \leq \frac{\alpha}{2R^2 L}.$$

Hence, From Lemma 3, we have that $g_{x_{\hat{i}}}(x)$ is $\frac{\alpha}{2}$ strongly convex at $x$, for all $\hat{i} \in I$:

$$v^\top \nabla^2 g_{x_{\hat{i}}}(x)v \geq \frac{\alpha}{2} \|v\|_{\mathcal{L}^c}^2. \tag{55}$$

Finally combining this with equation 54 we conclude

$$D^{2,l}g(x)[v,v], D^{2,r}g(x)[v,v] \geq \frac{\alpha}{2} \|v\|_{\mathcal{L}^c}^2.$$

$\square$

Next, we state the proof of Theorem 4.

*Proof of Theorem 4.* Consider the solution $\tilde{\mathcal{I}} = \left( \tilde{\mathbf{r}}, \tilde{\mathbf{v}}, \tilde{\boldsymbol{\Sigma}} \right)$ where $\forall i \in [N]$

$$\tilde{\Sigma}_{x_i} = \nabla^2 f(x_i),$$
$$\tilde{v}_{x_i} = \nabla f(x_i),$$
$$\tilde{r}_{x_i} = f(x_i).$$

First note that from Lemma 1 we get $f_{x_0}(x) + \frac{1}{6}\|x - x_0\|^3 \leq f(x)$, which implies $g_{x_i}^{(\tilde{\mathcal{I}})}(x_j) + \frac{15L}{96}\|x_j - x_i\|^3 \leq r_{x_j}$ for the above choice for $\tilde{\mathcal{I}}$. Moreover, $r_{x_0} \leq f(x_0) \leq C^2 \leq C_0$, and from $\tilde{c}_1$ Lipschitz and $\tilde{c}_2$ gradient Lipschitz conditions on $f$, we get $\forall i, \|\tilde{v}_{x_i}\| \leq \tilde{c}_1, \forall i, \tilde{\Sigma}_{x_i} \preccurlyeq \tilde{c}_2 I$, and the $\|.\|_{\mathcal{L}^c} - \alpha$ strong convexity of $f$ shows that $\tilde{\mathcal{I}}$ satisfies the condition $v^\top \Sigma_{x_i} v \geq \alpha, \forall v \in \mathcal{C}, \forall i$. Hence, $\tilde{\mathcal{I}}$ is feasible for the LP. In particular, note that we do not need the additional $L\bar{\epsilon}^3$ terms in the definition of $c_0, c_2, C_0$ to show the feasibility of $\tilde{\mathcal{I}}$ for the LP; these extra terms are only required for the third argument of Lemma 4 to show that not only $\tilde{\mathcal{I}}$ is feasible, but a ball around it is also feasible. We will prove that shortly. Next, from Lemma 2, we see that the maximum $\max_{i \in [N]} g_{x_i}^{\tilde{\mathcal{I}}}(x)$ at point $x \in \mathcal{X}$ is never achieved by far $x_j$'s from $x$, farther than $\|x_j - x\| \geq \gamma \left( \frac{\epsilon\sqrt{d}c_0}{L} \right)^{1/3}$, since the value of $g_{x_j}(x)$ is smaller than $g_{x_i}(x)$ for the element of the cover $x_i$ that is $\epsilon$ close to $x$. On the other hand, again from Lemma 2 for $x_i$ such that $\|x_i - x\| \leq \epsilon$ and any $x_j$ such that $\|x_j - x\| \leq \gamma \left( \frac{\epsilon\sqrt{d}c_0}{L} \right)^{1/3}$, we have

$$|g_{x_j}^{(\tilde{\mathcal{I}})}(x_i) - g_{x_j}^{(\tilde{\mathcal{I}})}(x)| \leq \gamma_2 \epsilon \sqrt{d}c_0,$$

and from LP feasibility

$$g_{x_j}^{(\tilde{\mathcal{I}})}(x_i) \leq r_{x_i}.$$

Therefore,

$$\max_{i \in [N]}|g_{x_i}^{(\tilde{\mathcal{I}})}(x)| \leq \max_{i \in [N]}|r_i| + \gamma_2 \epsilon \sqrt{d}c_0$$
$$= \max_{i \in [N]}|f(x_i)| + \gamma_2 \epsilon \sqrt{d}c_0$$
$$\leq C^2 + \gamma_2 \epsilon \sqrt{d}c_0.$$

Therefore, the optimal solution $\mathcal{I}^*$ should satisfy $\max_{i \in [N]}|g_{x_i}^{(\mathcal{I}^*)}(x)| \leq C^2 + \gamma_2 \epsilon \sqrt{d}c_0$ which proves the first argument equation 1. Finally, combining Lemmas 14 and 15 we get the $\alpha/2$ shows strong convexity of $g^{(\mathcal{I})}$ with respect to $\|.\|_{\mathcal{L}^c}$ for argument equation 2.

Next we show the third argument; note that $f$ satisfies a slightly stronger inequality compared to the first condition of the LP equation 9, namely

$$f(x_i) + \langle \nabla f(x_i), x_j - x_i \rangle + \frac{1}{2}(x_j - x_i)^\top \nabla^2 f(x_i)(x_j - x_i) - \frac{L}{3}\|x_j - x_i\|^3 \quad (56)$$

$$+ \left( \frac{L}{6} - \frac{L}{96} \right)\|x_j - x_i\|^3 + \frac{L}{96}\|x_j - x_i\|^3 \leq f(x_j), \quad (57)$$

or, since we constructed instance $\tilde{\mathcal{I}}$ from $f$,

$$g_{x_i}^{(\tilde{\mathcal{I}})}(x_j) + \frac{15L}{96}\|x_j - x_i\|^3 + \frac{L}{96}\|x_j - x_i\|^3 \leq f(x_j). \quad (58)$$

But if $\left\| \Sigma - \nabla^2 f(x_i) \right\| \leq \frac{L\bar{\epsilon}}{144} \leq \frac{L}{144}\|x_j - x_i\|$, then

$$\frac{1}{2}|(x_j - x_i)^\top \nabla^2 f(x_i)(x_j - x_i) - (x_j - x_i)^\top \Sigma(x_j - x_i)| \leq \frac{1}{2}\left\| (x_j - x_i)(x_j - x_i)^\top \right\|_F \left\| \nabla^2 f(x_i) - \Sigma \right\|_F$$
$$\leq \frac{1}{2}\|x_j - x_i\|^2 \left\| \nabla^2 f(x_i) - \Sigma \right\|_F$$
$$\leq \frac{L}{288}\|x_j - x_i\|^3.$$

Given $\|\nabla f(x_i) - v\| \leq \frac{L\bar{\epsilon}^2}{288} \leq \frac{L}{288}\|x_j - x_i\|^2$, we get

$$|\langle \nabla f(x_i), x_j - x_i \rangle - \langle v, x_j - x_i \rangle| \leq \|\nabla f(x_i) - v\| \, \|x_j - x_i\| \leq \frac{L}{288}] \|x_i - x_j\|^3.$$

Finally under $|f(x_i) - r| \leq \frac{L}{288}\bar{\epsilon}^3 \leq \frac{L}{288}\|x_j - x_i\|^3$. Hence, if we assume $\left\|\mathcal{I} - \tilde{\mathcal{I}}\right\| \leq \frac{L}{288}\bar{\epsilon}^3$, then combining the above Equations we get

$$|g_{x_i}^{(\mathcal{I})}(x_j) - g_{x_i}^{(\tilde{\mathcal{I}})}(x_j)| = |g_{x_i}^{(\mathcal{I})}(x_j) - f_{x_i}(x_j)| \leq \frac{L}{96}\|x_j - x_i\|^3.$$

But plugging this into equation 58

$$g_{x_i}^{(\mathcal{I})}(x_j) + \frac{15L}{96}\|x_j - x_i\|^3 \leq f(x_j), \tag{59}$$

Finally note that $\left\|\mathcal{I} - \tilde{\mathcal{I}}\right\| \leq \frac{L}{288}\bar{\epsilon}^3$ also implies $\forall i \in [N]$:

$$|r_{x_i}| \leq |r_{x_i} - \tilde{r}_{x_i}| + |\tilde{r}_{x_i}| \leq C^2 + L\bar{\epsilon}^3,$$
$$\|v_{x_i}\| \leq \|\tilde{v}_{x_i}\|_\infty + \|v_{x_i} - \tilde{v}_{x_i}\| \leq \tilde{c}_1 + L\bar{\epsilon}^3,$$
$$\Sigma_{x_i} \preccurlyeq \left\|\Sigma_{x_i} - \tilde{\Sigma}_{x_i}\right\| I + \tilde{\Sigma}_{x_i} \preccurlyeq \left(\tilde{c}_2 + L\bar{\epsilon}^3\right) I.$$

Therefore, $\tilde{\mathcal{I}}$ is still feasible for the program equation 9 with our choice of parameters $c_0, c_2, C_0$ here. Hence, we conclude

$$B_{L\bar{\epsilon}^3/288}(\tilde{\mathcal{I}}) \subseteq P_{\mathcal{I}} \subseteq B_{2\sqrt{(N+1)C_0{}^2 + Nd(c_0^2 + c_2^2)}}(\tilde{\mathcal{I}}).$$

Finally note that for arbitrary $\mathcal{I} \in \mathcal{P}_{\mathcal{I}}$ which satisfies the conditions in LP equation 9, we have

$$\|\mathcal{I}\|^2 \leq r^2 + \sum_i |r_{x_i}|^2 + \|v_{x_i}\|^2 + \|\Sigma_{x_i}\|^2$$
$$\leq (N+1)C_0{}^2 + Ndc_0^2 + Ndc_2^2,$$

which implies

$$P_{\mathcal{I}} \subseteq B_{2\sqrt{(N+1)C_0{}^2 + Nd(c_0^2 + c_2^2)}}(\tilde{\mathcal{I}}).$$

$\square$

## F.5 Proof of Theorem 5

Consider the random distribution in Theorem 1.2 of Bhattiprolu et al. (2021). Then from property (3), there exists a unit direction $v$ with $\|v\|_{\mathcal{L}} \leq \frac{1}{d^{1-\epsilon}}$. Then we claim that $\|v\|_{\mathcal{L}^c} \leq \frac{1}{d^{1-\epsilon}}$. This is because $\|v\|_{\mathcal{L}} = \sup_{\|w\|_{\mathcal{L}^c}} \langle v, w \rangle \geq \langle v, \frac{v}{\|v\|_{\mathcal{L}^c}} \rangle = \frac{1}{\|v\|_{\mathcal{L}^c}}$. Hence, $\|v\|_{\mathcal{L}}^c \geq d^{1-\epsilon}$. Hence, for $\tilde{v} = \frac{v}{\|v\|_{\mathcal{L}^c}}$ we have $\|\tilde{v}\|_{\mathcal{L}^c} = 1$ and $\|\tilde{v}\| \leq \frac{1}{d^{1-\epsilon}}$.

## G Strong convexity

Here we show that a lower bound on the second derivative implies strong convexity with respect to arbitrary norms.

**Lemma 15** (Lower bound on second derivative $\rightarrow$ strong convexity). *Suppose for convex function $g : \mathcal{X} \rightarrow \mathbb{R}$ which is second order continuously differentiable except in a finite number of points in which it is only left or right second order differentiable. Suppose the second left or right derivatives in arbitrary direction $v$, which we denote by $D^{2,l}g(x)[v,v], D^{2,r}g(x)[v,v]$ respectively, are at least $\alpha \|v\|_{\mathcal{L}^c}^2$. Then, $g$ is strongly convex with respect to $\|.\|_{\mathcal{L}}^c$, namely for any $x, y \in \mathcal{X}$ and any subgradient $v_x$ of $f$ at point $x$:*

$$f(y) \geq f(x) + \langle \nabla f(x), y - x \rangle + \alpha \|y - x\|_{\mathcal{L}^c}^2.$$

*Proof.* Without loss of generality assume $\|y - x\|_{\mathcal{L}^c} = 1$ and define the one variable function $h(t) : [0, 1] \to \mathbb{R}$: $h(t) = g(x + t(y - x))$, and let $0 \leq t_1 \leq t_2 \leq \cdots \leq t_k \leq 1$ are the non-differentiable points of $h(t)$ on $[0, 1]$, which we know are finite from our assumption. But from differentiability of $h$ between these points, we can write (define $t_0 = 0, t_{k+1} = 1$)

$$f(y) = g(1) = \sum_{i=1}^{k} \int_{t_i}^{t_{i+1}} g'(t)dt, \tag{60}$$

where for the integral in $[t_i, t_{i+1}]$ by $h'(t_i)$ and $h'(t_{i+1})$ we mean the right derivative $h'^r(t_i)$ and left derivative $h'^l(t_{i+1})$, respectively. Now we show that for all $t \in [0, 1]$

$$h'^l(t), h'^r(t) \geq h'^r(0) + \alpha t. \tag{61}$$

We show this inductively for $t \in (t_i, t_{i+1})$ for $i = 0, \ldots, k$. Particularly, the induction argument for step $i$ is that for $t \in (t_i, t_{i+1})$, $h'(t) \geq \alpha t + h'^r(0)$, and $h'^l(t_{i+1}), h'^r(t_{i+1}) \geq h'^r(0) + t_{i+1}\alpha$. The base trivial since $h'^r(0) \geq h'^r(0)$. For the step of induction from $i - 1$ to $i$, we know

$$g'^r(t_i) \geq \alpha t_i. \tag{62}$$

Now for any $t \in (t_i, t_{i+1})$ we can write

$$h'(t) = \int_{t_i}^{t} h''(s)ds \geq \alpha(t - t_i), \tag{63}$$

and particularly for $t_{i+1}$:

$$h'^l(t_{i+1}) = \int_{t_i}^{t_{i+1}} h''(s)ds \geq \alpha(t_{i+1} - t_i). \tag{64}$$

On the other hand, from the convexity of $g$,

$$h'^l(t_{i+1}) \leq h'^r(t_{i+1}). \tag{65}$$

Combining equation 63 equation 64 equation 65 with equation 62 completes the setp of induction.

Finally combining equation 61 with equation 60 and noting the fact that for any subgradient $v$ at point $x$, $\langle v, y - x, \leq \rangle h'^r(0)$,

$$f(y) \geq h'^r(0) + \int_0^1 \alpha t dt \geq h'^r(0) + \frac{\alpha}{2},$$

which completes the proof. $\qquad\square$

