# OpenReview forum: "Computing Optimal Regularizers for Online Linear Optimization"
_ICLR.cc/2025/Conference — ICLR 2025 Conference Withdrawn Submission_

### Official Review · Reviewer_tVAu · 2024-10-27

**Soundness:** 2
**Presentation:** 2
**Contribution:** 2
**Rating:** 3
**Confidence:** 4

**Summary:**

This paper investigates Follow-the-Regularized-Leader (FTRL) algorithms for online linear optimization, and aims to find the best regularizer for FTRL that is able to achieve the optimal constant factor in regret bounds. To this end, the authors propose a new algorithm that approximates the best regularizer by solving a set of convex programs.

**Strengths:**

- The authors have presented a detailed survy on related work in online linear optimization and FTRL methods.
- This paper is primarily theoretical and provides a complete proof of its findings.

**Weaknesses:**

- The writing and organization of this work require further revision. I suggest that the authors summarize their contributions and comparisons with existing methods in a table. Additionally, the authors should clearly outline their method in the form of an algorithm.
- I suggest that the author provide a high-level outline of the proof for Theorem 2, as the current version is overly hard to follow and redundant.

**Questions:**

- The core of this work is to achieve optimality of the constant term in the regret bound of the FTRL method by searching for the optimal regularization term. To accomplish this, the proposed method requires solving a set of convex programs, with a runtime that scales exponentially with $d$. This raises the question of whether it is justified to incur an exponentially dependent runtime solely to optimize the constant term. Therefore, the authors should emphasize the necessity of their research.

- Although I understand that this work primarily focuses on theoretical aspects, could the authors provide some experiments (even simulations) to illustrate the necessity of reducing the constant dependence in the regret bound of the FTRL method?

---

### Official Review · Reviewer_e4gf · 2024-10-29

**Soundness:** 2
**Presentation:** 1
**Contribution:** 2
**Rating:** 3
**Confidence:** 3

**Summary:**

Motivated by the fact that different regularizers could lead to different dependence of the regret on dimensionality when applying follow-the-regularized-leader (FTRL)  in online linear optimization (OLO), this paper investigates the problem of finding the optimal regularizer for FTRL in each specific OLO problem such that the dependence of the regret on dimensionality cannot be improved up to constant factors. Specifically, the authors prove the existence of such an optimal regularizer, which improves the existing result by an $\log T$ factor. Moreover, the authors propose a convex program for computing this ideal regularizer (in an approximated way).

**Strengths:**

1) In terms of the existence of the optimal regularizer, previous studies only prove that there is always an instance of FTRL with $O(Rate(\mathcal{X},\mathcal{L})(\log T)\sqrt{T})$. By contrast, in this paper, the authors show that there exists a regularizer that can reduce the regret of FTRL to $O(Rate(\mathcal{X},\mathcal{L})\sqrt{T})$.
2) The authors propose the first method for computing this ideal regularizer (in an approximated way), which is new to me. Moreover, from the existence of optimal regularizer (i.e., Theorem 2) to the final computable method, they actually provide an extension to the existence of optimal smooth regularizer (i.e., Theorem 3) and the corresponding approximation (Lemma 1), which are also new to me.

**Weaknesses:**

1) Although I agree with the significance of achieving the optimal regret for FTRL in specific OLO problems, the complexity of the proposed method seems too high, i.e., time and space exponential in $d$. Moreover, the authors do not provide any practical instance to show the usefulness of the proposed method, e.g., utilizing the proposed method to compute a regularizer of FTRL in some OLO problems, where people failed to find a regularizer with similar performance.
2) The writing of this paper needs some improvements, and the analysis is especially hard to follow. For example, this paper is more related to FTRL in the online learning setting, but the authors recall many related works about offline optimization, which seems to be of no benefit in understanding the technical details of this paper. In Section 4, when introducing the main results, the authors frequently refer the readers to the details in the appendix.

**Questions:**

1) Can the authors provide any practical instance to show the usefulness of the proposed method, e.g., utilizing the proposed method to compute a regularizer of FTRL in some OLO problems, where people failed to find a regularizer with similar performance?
2) It would be better if the authors could introduce more technical details about the existing studies of Srebro et al. (2011), which would make the reader easier to understand the improvement in the regret, and may also provide more insight about the technical details in the analysis of this paper.

---

### Official Review · Reviewer_cMfa · 2024-11-01

**Soundness:** 3
**Presentation:** 2
**Contribution:** 3
**Rating:** 6
**Confidence:** 3

**Summary:**

This work shows that for any online linear optimization problem with symmetric action and loss sets, there exists a regularizer $f_0$ s.t. running FTRL using $f_0$ achieves the minimax optimal regret of $O(\operatorname{Rate}(\mathcal{X}, \mathcal{L}) \sqrt{T})$, improving the results of [1] by a factor of $\log T$. Further, the authors give a construction method of such regularizer using Gaussian smoothing. The authors also show that for any given smooth regularizer $f$, it can be approximated using “quasi-quadratic” functions, without sacrificing the optimality of the regularizer.

[1] Srebro et al. On the universality of online mirror descent. NeurIPS, 2011.

**Strengths:**

1. Technical innovations: The result of the universal regularizer improving the previous result in [1]. The authors also provide construction methods for such universal regularizers.

**Weaknesses:**

Some parts of the work are a bit hard to follow. For instance,
1. In Theorem 1, does the author mean $ \sup _{x \in \mathcal{X}}|g(x)|=O\left(\operatorname{Rate}(\mathcal{X}, \mathcal{L})^2\right) $ by $ \sup _{x \in \mathcal{X}}|g|=O\left(\operatorname{Rate}(\mathcal{X}, \mathcal{L})^2\right) $?
2. In Theorem 1, it would be better if the definition or description of the “membership oracle” is introduced first, before stating “given access to a membership oracle”.
3. Also in Theorem 1, what does it mean by “a cutting-plane algorithm that runs FTRL with regularizer $g$”? Are the authors implying that the cutting-plane algorithm can be used to solve Eq. (1)?
4. The descriptions on Line 291 – Line 297 are a bit confusing. On Line 293, the authors say that it is important to make the resulting regularizer have “bounded gradients”. However, on Line 297, it turns out that the authors use “Gaussian smoothing” to ensure the “smooth gradients” instead of “bounded gradients”.
5. On Line 299, what does it mean by “optimize over the space of smooth convex functions” and why do we need to consider “optimizing over the space of smooth convex functions”?
6. Some theorems seem to lack sufficient discussions and explanations (say, Theorem 4). I would suggest the authors give more discussions on such theorems.
7. I understand that the ultimate goal in this work is to derive the desired regularizer in the general case. However, I think it would benefit the audience if the authors could demonstrate some special cases where the method in this work can derive some appealing regularizers that are previously unknown.

**Questions:**

1. After the optimal regularizer is computed, can we solve the update of FTRL in Eq. (1) in a computationally efficient manner?
2. What can we say in the case of bandit feedback?

---

### Note · Authors · 2024-12-02

I have read and agree with the venue's withdrawal policy on behalf of myself and my co-authors.